

# Interspecific variation in tropical tree height and crown allometries in relation to life history traits

Isabel Martínez Cano[1*], Helene C. Muller-Landau[2], S. Joseph Wright[2], Stephanie A. Bohlman[2,3] and Stephen W. Pacala[1]

[1]Department of Ecology and Evolutionary Biology, Princeton University, Princeton, NJ 08544, USA
[2]Smithsonian Tropical Research Institute, Box 0843-03092, Balboa, Ancón, Panamá
[3]School of Forest Resources and Conservation, University of Florida, Gainesville, Florida 32601, USA

*Correspondence to*: Isabel Martínez Cano (isamcano@gmail.com)

**Abstract.** Tree allometric relationships are widely employed to estimate forest biomass and production, and are basic
building blocks of dynamic vegetation models. In tropical forests, allometric relationships are often modeled by fitting scale-invariant power functions to pooled data from multiple species, an approach that fails to reflect finite size effects at the smallest and largest sizes, and that ignores interspecific differences in allometry. Here, we analyzed allometric relationships of tree height (9884 individuals) and crown area (2425) with trunk diameter using species-specific morphological and life history data of 162 species from Barro Colorado Island, Panamá. We fit nonlinear, hierarchical models informed by species
traits and assessed the performance of three alternative functional forms: the scale-invariant power function, and the saturating Weibull and generalized Michaelis-Menten (gMM) functions. The relationship of tree height with trunk diameter was best fit by a saturating gMM model in which variation in allometric parameters was related to interspecific differences in sapling growth rates, a measure of regeneration light demand. Light-demanding species attained taller heights at comparatively smaller diameters as juveniles and had shorter asymptotic heights at larger diameters as adults. The
relationship of crown area with trunk diameter was best fit by a power function model incorporating a weak positive relationship between crown area and species-specific wood density. The use of saturating functional forms and the incorporation of functional traits in tree allometric models is a promising approach to improve estimates of forest biomass and productivity. Our results provide an improved basis for parameterizing tropical tree functional types in vegetation models.

## 1 Introduction

Allometric scaling encapsulates emergent evolutionary and physical constraints on plant morphology and performance that vary with organism size (Niklas, 1994). Allometric relationships show high predictive ability and are widely employed to estimate forest carbon biomass and primary production from forest inventory data (e.g., Chave et al., 2014; Goodman et al., 2014). Allometric functions constitute building blocks of more complex, mechanistic forest models, including the vegetation
modules of state of the art Earth System Models (e.g., Weng et al., 2015). These functions provide a basic template to model



carbon allocation and tree growth, and differences in allometric parameters can be used to represent different species or plant functional types (Pacala et al., 1996). However, allometric relationships of tropical trees remain poorly documented when compared to temperate and boreal forest ecosystems (Houghton, 2005; Hunter et al., 2013), even though tropical forests account for a disproportionate share of forest carbon stocks and fluxes (~50% of the terrestrial carbon sink, and ~33% of

annual NPP; Chapin, 2011; Pan et al., 2013). Alternative choices of tree allometric equations contribute to the wide variability in biomass and productivity estimates in the literature, and to the large uncertainty surrounding the response of these ecosystems to a warmer and dryer climate (Bonan, 2008).

Power functions are widely used to describe allometric scaling of tree dimensions with trunk diameter, despite the known limitations of their underlying assumption of scale-invariance of tree morphology (Shinozaki et al., 1964b, a; Niklas,

1994). The adoption of power function scaling is particularly problematic at both extremes of the tree size range (Enquist and Bentley, 2012). Power functions fail to capture the allometries of the smallest and largest individuals, generally underestimating dimensions of seedlings and saplings, and overestimating the size of large trees (e.g., Fayolle et al., 2016; Ledo et al., 2016). This suggests the need for alternative functional forms to represent life history heterogeneities and the physical constraints that set maximum tree sizes (Koch et al., 2004; Bonan, 2008; Goodman et al., 2014). Indeed, the

inclusion of a saturating relationship for tree scaling has proved important to reproduce realistic dynamics in vegetation models (Weng et al., 2015).

Allometric studies of tropical trees have highlighted differences in growth and morphology that define distinct life history strategies (Clark and Clark, 1992). These differences are known to promote species coexistence and play a key role in successional trajectories (Wright, 2002; Chazdon, 2014; Falster et al., 2017). Approaches pooling data across species

inherently fail to recognize species heterogeneity and limit the potential to identify and define plant functional groups. Pooling data across species also tends to over represent locally abundant species unless appropriate methods like hierarchical models are employed to account for unbalanced sampling. Differences among species are in most cases systematic, suggesting that they reflect life-history, mechanical, or phylogenetic constraints that cannot be interpreted as random variability or changes in environmental conditions alone (Westoby et al., 2002; Adler et al., 2014). This suggests that

hierarchical approaches based on functional traits might prove useful in understanding interspecific variation in tree allometry (Dietze et al., 2008; Iida et al., 2011).

Here, we present a quantitative approach to characterize allometric relationships for tree height and crown area and their interspecific variation in tropical forests, and apply it to a large dataset for a single site. Our overall objective was to develop models informed by functional traits to capture interspecific variation in the allometric scaling of tropical trees and

provide a better template for the estimation and modeling of forest biomass and ecosystem fluxes. We address three specific questions: (i) How is interspecific variability in allometric scaling of tree height and crown area related to tree species functional traits, in particular wood density and measures of shade tolerance? (ii) How do power functions compare with various asymptotic functions in representing these allometric relationships? (iii) How does the choice of alternative tree height scaling relationships affect the estimation of aboveground biomass? To answer these questions, we fitted allometric




models whose parameters were related to species-specific functional traits under a Bayesian hierarchical framework, taking advantage of long-term, high-quality data from Barro Colorado, Panamá. This approach allowed us to characterize different sources of variability, from individual species to the community level, and to assess the relative merits of different functional forms.

## 2 Methods

### 2.1 Study site

The Barro Colorado Nature Monument is a protected area in central Panama consisting of Barro Colorado Island (BCI) and peninsulas on the surrounding mainland (Leigh, 1999). The vegetation is moist tropical forest. Annual rainfall averages 2657 mm (years 1926 to 2017), with a 4-month dry season approximately from mid-December to mid-April (Paton, 2018). The forest dynamics plot on BCI is a 50-ha area (1000 m x 500 m) in which all trees with trunk diameter of 1 cm or larger have been measured, mapped, tagged, and identified to species in regular censuses since the early 1980s (Hubbell, 1983; Condit, 1998; Hubbell et al., 1999; Leigh, 1999; Hubbell et al., 2005). The plot is mostly old-growth forest of 400 years or older (Piperno, 1990), with the exception of a small area of old secondary forest ~127-137 years old in the central part of the northern edge of the plot (Mascaro et al., 2011). The Gigante peninsula on the nearby mainland is covered by secondary forest ranging from 100 to perhaps 300 years old (Denslow and Guzman G., 2000).

### 2.2 Allometric data

The allometric data consist of measurements of trunk diameter at 1.3 m height or above buttresses, $D$ (cm), tree height, $H$ (m), and crown area, $C$ (m$^2$). We used a compilation of seven datasets collected in the BCI 50 ha plot and one dataset from the adjacent Gigante peninsula. The datasets cover different size classes and combine measurements made with different methods (Table S1). In the case of crown area measurements derived from aerial photographs, we restricted our analyses to individuals whose crowns were fully sun-exposed as assessed by ground-based observers, to avoid crown area underestimation. We included only species with at least five individual measurements of either $H$ or $C$ and having data for the three trait covariates (see below), which resulted in a pool of 162 species including 9,884 trees for height allometries and 2,425 trees for crown area allometries.

### 2.3 Species traits

We considered three species-level covariates to assess whether ecological traits can explain interspecific variability in allometric scaling: the structural trait of wood density, and two demographically based indicators of shade-tolerance. Species-specific wood density values (dry matter weight per unit of fresh volume, g cm$^{-3}$) –technically wood specific gravity (Williamson and Wiemann, 2010)– were based on measurements taken in central Panama (Wright et al., 2010). The two





shade-tolerance indicators were the rates of mean sapling diameter growth and of sapling mortality estimated by Condit et al. (2006) at BCI using the five-year census data between 1982 and 2005. Sapling relative growth rates (% yr$^{-1}$) were based on diameter increments for individuals between 19 and 49 mm in diameter. Annual mortality rates (% yr$^{-1}$) were also based on the monitoring of tagged individuals, but including saplings with diameters between 10 and 99 mm (Condit et al., 2006).

5    **2.4 Statistical analyses**

We adopted a Hierarchical Bayesian (HB) approach to analyze allometric relationships of tree height and crown area with trunk diameter (Dietze et al., 2008; Price et al., 2009; Iida et al., 2011). The HB approach provides several advantages over classic analytical frameworks (Cressie et al., 2009; Gelman, 2014), starting with the easy accommodation of complex data structures and process models. In the current context, the HB framework allowed us to simultaneously estimate (i) community-level allometries that best represent an average species in the community, (ii) species-specific allometries that capture interspecific variation, and (iii) general relationships of species-specific allometric parameters to functional traits. The estimation of general relationships is improved by properly weighting species-specific estimates, whereas species-level estimates for rare species benefit from borrowing strength from the community-level relationship. The latter aspect reduces the negative impact of outlying observations and allows inference in data-poor species, of which there are many in hyperdiverse systems like tropical forests. Another important advantage of the HB approach is that it can easily handle nonlinear models, allowing us to extend the analysis beyond power functions (which are typically fitted through linear regressions on log-transformed data) to other functional relationships. Finally, HB allowed us to assess the effects of functional traits on tree allometries by explicitly including a relationship between species-specific allometric parameters, life history traits and biomechanical constraints.

20    **2.4.1 Model specification**

Bayesian hierarchical models have three components (Cressie et al., 2009): (*i*) a data model linking model predictions with observed data, (*ii*) a process model providing a mathematical description of the mechanisms underlying the patterns of interest, and (*iii*) a parameter model that incorporates prior information about parameter values available before the analysis. For the data model, we assumed a Gaussian likelihood for the natural logarithm of the response variable, $y_{s[i]}$, which was either tree height (*H*, m) or crown area (*C*, m$^2$) for each individual *i* in species *s*;

$$\log y_{s[i]} \sim Gaussian\left(f\left(D_{s[i]}, \theta_s\right), \sigma_v\right) \qquad (1)$$

where the process model, $f(\cdot)$, predicts expected tree height or crown area from observed trunk diameter, *D* (cm), and the vector of species-specific parameters, $\theta_s = \{a, b, k\}$, and the standard deviation $\sigma_v$ captures deviations between model predictions and observed data.




We considered three functional forms of varying complexity for the process model, representing alternative hypotheses about allometric scaling. Our simplest model was the power function model, which presumes scale invariance of tree morphology with trunk diameter. We also tested two models that are nonlinear in the logarithmic scale, thereby allowing for a curvature in scaling (Thomas, 1996); a generalized Michaelis-Menten (gMM) and a rescaled Weibull function (the

cumulative Weibull distribution rescaled to extend from 0 to $a$ rather than 0 to 1). These functions were chosen because they are always non-decreasing and allow for finite constraints on maximum tree dimensions, with both equations featuring a saturating relationship between tree dimensions and trunk size. The equations for these models are

| | | | |
|---|---|---|---|
| *Power* | | $y = aD^b$ | (2) |
| *Generalized Michaelis-Menten (gMM)* | | $y = \frac{aD^b}{k+D^b}$ | (3) |
| *Weibull* | | $y = a\left(1 - exp(-bD^k)\right)$ | (4) |

Preliminary analyses evaluated additional saturating functional forms, including the Gompertz and logistic, and found that they produced inferior fits, in agreement with previous studies of tree height and crown area (Feldpausch et al., 2011; Banin

et al., 2012; Ledo et al., 2016).

The effect of the functional traits was evaluated by adding an additional layer to the process model to accommodate interspecific differences in model parameters. Each species-specific parameter, $\theta_s$, of each allometric relationship (i.e., $a$, $b$, or $k$ in equations 2, 3, or 4) was modeled as a univariate linear function of one of the covariate traits of interest, $T_s$ (i.e., wood density, sapling mortality rate, or sapling growth rate):

$$\theta_s \sim Gaussian(\alpha_\theta + \beta_\theta T_s, \sigma_\theta) \qquad (5)$$

Deviations from the linear relationship were assumed to follow a normal distribution with a community-level standard deviation $\sigma_\theta$. Each covariate was centered and scaled to mean zero and unit variance before the analysis. As a consequence,

the intercept of the linear relationship among parameter values and species-specific traits, $\alpha_\theta$, provides an estimate of the across-species mean, while the slope $\beta_\theta$ gives the expected effect of an increase in one unit standard deviation for each covariate (Gelman, 2014). We compared models including individual functional traits with models lacking covariates, that is, models in which variation among allometric scaling parameters is assumed to be random (i.e., equivalent to setting $\beta_\theta$ to zero). Each trait model modeled all parameters as functions of the same focal trait; thus, the trait models had twice the

number of community-level parameters as corresponding models lacking covariates.

The model was completed with the specification of noninformative prior distributions in the parameter model. We assumed independent normal priors for the vector of community- and species-level parameters. The distribution of $\alpha_\theta$ and $\beta_\theta$ acts as a prior in the characterization of allometries for each species. Assuming a multivariate normal prior for model parameters did





not significantly alter the main results. We assumed a half-Cauchy prior distribution for the observation variance and for the across-species variances of the parameters of the allometric models.

### 2.4.2 Model selection and inference

For both tree height and crown area, model selection and inference involved the assessment of 12 different model formulations resulting from all combinations of the three process models (power, generalized Michaelis-Menten and Weibull), and the four possibilities for functional traits (no trait, wood density, sapling growth, and sapling mortality). Alternative models were fitted using Markov chain Monte Carlo (MCMC) methods (Gelman, 2014). Inference was based on 5,000 posterior samples following 10,000 burn-in iterations for four parallel chains, which allowed us to check convergence using the potential scale reduction statistic together with estimates of effective sample size (Gelman and Rubin, 1992). Based on the posterior distribution of the deviance, we calculated the Watanabe (2013) information criterion (WAIC) to rank alternative models in terms of a balance between predictive ability and model complexity (Hooten and Hobbs, 2015). Models were fitted in *Stan* (Stan Development Team, 2016), a statistical software package to conduct Bayesian analyses (code provided in Appendix S2).

Posterior samples were used to characterize the distribution of parameters and to project estimation uncertainty to model-based estimates. We report central, 90% posterior intervals both for parameter estimates and for model-based predictions of tree height and crown area for selected trunk diameter values. We further provide unbiased community-level models for estimating (untransformed) height and crown area from trunk diameter. These models were corrected for the bias introduced by back-transformation of log-transformed predictions; the correction involves multiplying predicted values by $\exp(\sigma_v^2/2)$, where $\sigma_v$ is the residual standard deviation of the fitted model for the log-transformed variable (Sprugel, 1983).

### 2.4.3 Implications for biomass estimates

Finally, we derived estimates of oven-dry aboveground biomass, *AGB* (kg dry mass) from measured trunk diameters and our estimated heights, using a general tropical tree allometric model fitted by Chave et al. (2014):

$$AGB = 0.0559 \times (\rho D^2 H) \qquad (6)$$

where $\rho$ is wood density (g cm$^{-3}$), $D$ (cm) is trunk diameter and $H$ (m) is tree height. We first compared *AGB* estimates based on measured tree heights with the corresponding *AGB* estimated using community-level, model-based predictions of tree height from alternative functional forms (i.e., power vs. gMM), and evaluated how these differences varied with tree diameter. Then, we estimated total *AGB* in the 50 ha BCI census plot by summing over individual tree estimates of AGB, using individual *D* measurements from the 2010 census (Hubbell et al., 2005). In this second round of comparisons, we explored the impact of species-specific differences in height allometric scaling by comparing *AGB* estimates based alternatively on community- or species-level height predictions for both power and gMM functions. Species-specific height



predictions were available only for the 162 species included in the main analysis, so we used community-level predictions for other species. For those species for which species-specific wood densities were not available, we substituted the average over species for which values were available ($\rho$ = 0.5304 g/cm3, Wright et al., 2010).

## 3 Results

Trees in our dataset varied over three orders of magnitude in trunk diameter (0.33–247.70 cm), two orders of magnitude in tree height (0.55–57.40 m) and five orders of magnitude in crown area (0.0039–1404.2 m$^2$). Observations were unevenly distributed across species, in large part in parallel with the variation in abundance, with a median [range] of 34 [5–674] trees per species for tree height and 7 [3–139] for crown area. The hierarchical models accounted for this unbalanced design and provided reasonable fits in all species for all model combinations, with no apparent pattern remaining in the residuals (Figs.

S1-S2, Table 1). The goodness of fit of all candidate models was in general high, with coefficients of determination ($r^2$) between 0.909 and 0.943. Differences in the accuracy and precision of predictions resulted nonetheless in a clear ranking among alternative models according to WAIC (Table 1).

### 3.1 Tree height allometry

The best tree height model combined a generalized Michaelis-Menten (gMM) function (Fig. 1a) with dependence of species-

specific parameters on sapling growth rates. At the community level, the best model for predicting tree height, $H$ (m), from trunk diameter, $D$ (cm) in the absence of information on species-level covariates was

$$H = \frac{58.0 D^{0.73}}{21.8 + D^{0.73}} \qquad (7)$$

This equation incorporates the bias correction for the back-transformation from log $H$ based on the estimate of $\sigma_v$ = 0.181 [0.179, 0.183]$_{90\%}$. The parameter values with their 90% posterior central intervals are asymptote $a$ = 57.1 [54.5, 60.0]$_{90\%}$, before bias correction, exponent $b$ = 0.73 [0.72, 0.75]$_{90\%}$, and half-saturation parameter $k$= 21.79 [20.70, 22.89]$_{90\%}$.

Individual species showed considerable variation in their height allometries, variation that was explained to a large extent by sapling growth rate (Fig. 2). Parameters $a$ and $k$ declined with sapling growth rate, while $b$ increased (Fig. 2; $r^2$ = 0.42, 0.14

and 0.16 for relationships with $a$, $b$ and $k$, respectively, with the natural logarithm of sapling relative growth rate). As a consequence, fast-growing species attain taller heights at comparatively small diameters but have shorter asymptotic heights compared with slow-growing species (Fig. 2d). The second-best model included a generalized Michaelis-Menten function and no covariates, but was significantly worse in WAIC ($\Delta$WAIC=5.6). The third-best model, which incorporated a Weibull function and interspecific variation in sapling growth rates (Table 1) produced qualitatively the same results as the best



model. In general, all the saturating models (Weibull or gMM), regardless of covariate or not, had fairly similar log pointwise predictive density (ΔELP<10), whereas the power function models did much worse (ΔELP>500, ΔWAIC>1000).

### 3.2 Crown area allometry

The allometric scaling of crown area with trunk diameter showed no sign of saturation, and thus the power function model provided superior fits (Fig. 1b). At the community level, the best model for predicting crown area, $C$ (m$^2$), from trunk diameter, $D$ (cm) in the absence of information on species-level covariates was

$$C = 0.66D^{1.34} \qquad (8)$$

This equation incorporates the bias correction for the back-transformation from log $C$ based on $\sigma_v = 0.549$ [0.536, 0.563]$_{90\%}$. The parameter values with their 90% posterior central intervals are $a = 0.57$ [0.50, 0.63]$_{90\%}$ before bias correction, and exponent $b = 1.34$ [1.31-1.38]$_{90\%}$. This model does not have an asymptote. For the maximum trunk diameter in our dataset, $D_{max} = 250$ cm, we would expect a crown area close to $C_{max} = 1079$ [977, 1192]$_{90\%}$ m$^2$, corresponding to a radius of 18.5 m. Although the best model did not include a covariate, the model including wood density provided a competitive fit (ΔWAIC=0.6, Table 1), with a slight positive relationship between the intercept and wood density (Fig. 3, $r^2 = 0.02$ and 0.01 for the relationships presented). This suggests that species with high wood densities tend to have slightly broader crowns at all trunk sizes.

### 3.3 Consequences for AGB estimates

Individual tree aboveground biomass (*AGB*) estimates based on the power model were strongly upwardly biased for large trees relative to estimates based on measured heights, whereas AGB estimates based on the gMM height model were unbiased (Fig. 4). The power model estimates exceeded the gMM model estimates by ever larger proportions at higher trunk diameters, with an overestimate of 10% at $D = 66$ cm [52, 80]$_{90\%}$, reaching 59% [51, 67]$_{90\%}$ at $D = 250$ cm (Figs. 4 and 5). This difference in AGB estimates for large trees translated into substantial differences for whole-plot AGB. Estimates of total *AGB* in trees with $D \geq 1$ cm using the community-average power model ($H = 3.02D^{0.56}$) were 12.3% larger than those using the better-supported community-average gMM model (283 vs. 252 Mg dry matter ha$^{-1}$, Table 3). The incorporation of information about species identity reduces the difference between the models, with the power model estimate exceeding the gMM model estimate by only 4.5% (276 vs. 264 Mg ha$^{-1}$). As expected, deviations between estimates based on the power and the gMM models were more pronounced for large diameter classes (Table 3).



## 4 Discussion

Tree allometric relationships are widely employed to estimate forest biomass and production, and they are also the basic building blocks guiding the development and validation of dynamic vegetation models. In tropical forests, the high diversity of trees makes it difficult to collect sufficient data to characterize species-specific allometric scaling relationships for any

substantial fraction of the flora. Here, we analyzed tree allometric scaling at Barro Colorado Island using Bayesian nonlinear hierarchical models. Taking advantage of a large, species-specific dataset of tree morphology and functional traits, we uncovered contrasting allometric relationships for the scaling of tree height and crown area with trunk diameter, with distinct associations with ecological traits.

### 4.1 Tree height allometry

Our analysis supported a saturating relationship between tree height and trunk diameter, consistent with theory (Falster and Westoby, 2003; Niklas, 2007) and with previous studies in tropical forests (Thomas, 1996; Bullock, 2000; Banin et al., 2012; Feldpausch et al., 2012; Molto et al., 2014; Fayolle et al., 2016; Ledo et al., 2016) and other forest biomes (e.g., Canham et al., 1994). The deceleration of height with respect to trunk diameter has been explained by multiple non-exclusive mechanisms including mechanical resistance (e.g., McMahon, 1973), growth and hydraulic constraints (e.g., Niklas and

Spatz, 2004), and asymmetric competition for light (e.g., Iwasa et al., 1985; Falster and Westoby, 2003; Bohlman and O'Brien, 2006). Past work suggests that mechanical resistance to self- or wind-loading cannot explain tree height allometries, as trees are generally much shorter for a given diameter than the limits based on static mechanical constraints (Niklas, 2007). Metabolic theories based on hydraulic constraints predict a constant logarithmic scaling between tree height and trunk diameter with an exponent close to 2/3 (Niklas and Spatz, 2004; West et al., 2009), inconsistent with our results.

Interspecific variation in tree height scaling parameters was associated with sapling growth rates, an indicator of shade-tolerance (Condit et al., 2006). In tropical forests, competition for light is traditionally invoked to explain the decreasing allocation of resources to tree height growth during ontogeny (Muller-Landau et al., 2006; Iida et al., 2011; Banin et al., 2012). The association between allometric parameters and regeneration ability spans a continuum of strategies. On the one hand are light-demanding tree species that attain larger heights at comparatively small trunk diameters in canopy gaps,

mature at relatively short sizes, and are eventually overtopped. On the other hand are shade-tolerant species whose saplings persist in the shaded understory, until they eventually reach taller heights in the canopy where they overgrow and suppress other trees (Bohlman and O'Brien, 2006; Bohlman and Pacala, 2012; Farrior et al., 2016). In old growth forests like BCI, shade-tolerant species dominate in numbers overall, whereas opportunistic light-demanding species thrive when tree-fall events open gaps in the canopy and constitute approximately half of canopy basal area (Bohlman, 2015; Farrior et al., 2016).

Our results show how variation in shade-tolerance aligns with differences in the parameters of saturating height allometric functions, and thereby provide a basis to define plant functional types representative of different gap-successional stages in tropical forests (Thomas, 1996; Falster et al., 2017).



## 4.2 Crown area allometry

Crown area presented a constant scaling with trunk diameter and no indication of saturation, even for the larger trees in our dataset. As a consequence, the model selection procedure favored power function models with estimates of the community-level exponent close to 4/3 ($b = 1.35$ [1.31, 1.38]$_{90\%}$). This result is consistent with previous analyses across large scale

environmental gradients reporting allometric exponents for crown area between 1.21 and 1.36 (Bohlman and O'Brien, 2006; Muller-Landau et al., 2006; Heineman et al., 2011; Antin et al., 2013; Blanchard et al., 2016). This large-scale consistency in community-level relationships emerges despite local variation among species, and suggests the operation of a general mechanism in the emergence of community-level allometric scaling in crown geometry (Farrior et al., 2016). The fitted community-level crown area exponent is consistent with predictions of 4/3 scaling by elastic similarity models describing

mechanical resistance to wind (McMahon, 1973), as well as by metabolic models invoking design constraints in transportation networks (West et al., 2009).

Our finding of high interspecific variation in the allometric scaling of crown geometry is consistent with previous studies (Iida et al., 2012; Blanchard et al., 2016). This interspecific variation has been linked with local differentiation and niche partitioning into canopy layers (Clark et al., 2008; Bohlman and Pacala, 2012). For instance, the crowns of subcanopy

trees are wider than those of tall-statured trees at BCI (Bohlman and O'Brien, 2006). Our analysis of crown area favored an allometric model lacking trait influences on species-specific parameters, although the model featuring a weak positive relationship between the intercept of the power function and the average wood density of each species also received considerable support. The estimated relationship of crown area to wood density is consistent with theory that high wood density enables more efficient horizontal crown expansion (Anten and Schieving, 2010). Using similar methods, both Iida et

al. (2012) and Francis et al. (2017) found a trend for trees with higher wood density to have larger crowns for a given diameter in Pasoh (Malaysia), although the relationship for crown width emerged only for trees smaller than 18 m. The lack of association for large trees at Pasoh might help explain the weak relationship found at BCI (i.e., one third of the trees included in our analysis were taller than 18 m), and it suggests that other factors can be important in shaping crown geometry in large trees, including resource partitioning within stands (Muller-Landau et al., 2006).

## 4.3 Implications for forest biomass estimation

Allometric models for individual trees remain the preferred method to estimate forest biomass and production at the stand level from plot data (Chave et al., 2014; Brienen et al., 2015), and provide a basic template to model carbon allocation and tree growth and competition in dynamic vegetation models. Whereas many models incorporate only trunk diameter (Brown, 1997), current state-of-the-art models typically include estimates of wood density and tree height as well (e.g., Chave et al.,

2014), and crown dimensions have also been incorporated in some models (Goodman et al., 2014; Ploton et al., 2016). Inclusion of height and/or crown dimensions in tree biomass models reduces errors in biomass estimates, especially for large trees, which contribute disproportionately to forest biomass and function (Lindenmayer et al., 2012).



Our results highlight the importance for biomass estimation of accounting for saturation in height-diameter allometries and interspecific variation in allometric parameters. If heights are not directly measured, any estimates of heights should be based on fitting saturating height functions to datasets with sufficient data for large trees to accurately capture the saturating component (Sullivan et al., 2018). The use of power function fits for heights leads to substantial overestimates of

biomass of large trees, which translates to substantial overestimates of stand-level biomass. The considerable heterogeneity among species in both tree height and crown area allometries presents another opportunity to improve estimates of forest biomass. The use of average allometric models that ignore changes in species composition can result in biased estimates of total biomass, reflecting the underlying nonlinearities of these relationships. At the same time, it is clearly impractical to develop species-specific allometries for every tropical tree species. The use of hierarchical models based on functional or

demographic traits provides a manageable option for incorporating and accounting for the diversity of allometric scaling relationships in biomass models. Ideally, such hierarchical models would be grounded in a mechanistic understanding of underlying tradeoffs on trait diversity (Falster et al., 2017).

## 4.4 Conclusions and directions for future research

Despite growing evidence highlighting the deceleration in tree height scaling (e.g., Thomas, 1996; Bullock, 2000; Banin et

al., 2012; Ledo et al., 2016), the power function remains the most commonly used model of tree height allometry in tropical forests (e.g., Antin et al., 2013; Goodman et al., 2014; Blanchard et al., 2016). Even studies featuring saturating relationships often fix the exponent of the gMM or Weibull functions to unity (e.g., Banin et al., 2012; Ledo et al., 2016), a value that our results show is inconsistent with the data. Our results favored the gMM function over the modified three-parameter Weibull previously proposed by Thomas (1996), suggesting it provides the required level of flexibility to accommodate changes in

tree height scaling during ontogeny. Three-parameter saturating models clearly outperform two-parameter power functions in large datasets containing data on many large individuals; however, the advantage in fit of the saturating models is often insufficient to compensate for the penalty of an extra parameter in the many cases in which smaller datasets or those with data for few larger individuals are analyzed in isolation (Thomas, 1996; Iida et al., 2011; Goodman et al., 2014). We recommend that future analyses of small datasets on tropical tree allometries be conducted in a Bayesian framework in

which prior data for larger datasets informs the choice of functional forms (i.e., restriction to saturating functions) and informs prior distributions on parameter values. Future studies can take advantage of our data, code, and results to constrain inferences on tree allometry using informative priors (e.g., Ellison, 2004).

The analysis of crown allometric scaling involved an unusually large dataset, yet remained limited by sample size, measurement difficulties, and failure to address other dimensions of crown size such as crown depth. The crown area dataset

was only one fifth the size of our tree height dataset, and only one fourteenth of the trees had trunk diameters greater than 100 cm. For ground-based data, which constituted the vast majority of our dataset, measurements of crown dimensions are more complicated and time-consuming than those of height, and we expect them to have higher measurement error. Aerial and even satellite imagery increasingly offers an alternative for precisely and accurately measuring crown areas of fully sun-





exposed trees, an alternative we took advantage of here. However, these methods do not enable crown area estimates for sub-canopy trees (but see e.g. Paris et al., 2016; Shendryk et al., 2016), which differ systematically in their crown allometries. Finally, we evaluated only crown area, even though crown depth and crown shape are also important for the estimation of tree biomass (Goodman et al., 2014; Ploton et al., 2016) and for characterizing tree life history strategies (Canham et al.,

1994; Bohlman and O'Brien, 2006). Despite these limitations, our analysis consistently favored crown area-DBH models without saturation and suggested a weak effect of species differences in wood density on the considerable interspecific variability in the scaling of crown size.

Besides the need to continue improving the characterization of variation in tree morphology, it is important to broaden the scope of allometric studies by complementing available data sets with ancillary data. There is room to continue

exploring and refining the relationship between allometric parameters and functional traits, especially in the case of demographic rates (Condit et al., 2006). The models presented here can be further extended by considering additional sources of heterogeneity in allometry, including environmental conditions, biogeographic region, and competitive neighborhood. Previous studies show that tropical tree height allometries vary with climate (Chave et al., 2014), topography (Ferry et al., 2010; Marshall et al., 2012), edaphic conditions (Aiba and Kitayama, 1999; Feldpausch et al., 2011), canopy

position (O'Brien et al., 1995; Thomas, 1996; Poorter et al., 2006), and light exposure (Rüger et al., 2012). Ideally, these factors would be incorporated not in a purely phenomenological manner, but informed by mechanistic models of underlying tradeoffs and alternative strategies (Dybzinski et al., 2011; Farrior et al., 2013).

Tree allometric functions are critical components of forest biomass estimates and of mechanistic models of forest structure and dynamics. Models reproducing the size structure of tropical forests are highly sensitive to crown allometry

parameters (Farrior et al., 2016), and the responses of vegetation models depend critically on the functional forms for tree allometry (Weng et al., 2017). Our results show that allometric models for tropical trees should incorporate saturating functions for tree height and interspecific variation in scaling parameters. In our analyses of data for over 10,000 tropical trees, tree height presented a saturating relationship with trunk diameter that was well-captured by the three-parameter generalized Michaelis-Menten or Weibull functions, whereas power function models exhibited systematic biases in tree

height predictions, especially for large trees. In contrast, our somewhat smaller dataset for crown area exhibited a constant scaling with stem size, in accordance with a power function. We observed extensive interspecific variability in allometric scaling, with this variability linked to shade-tolerance for height and with wood density for crown area. The relationship of tree allometric parameters with functional and demographic traits paves the way for the incorporation of compositional effects into estimates of forest biomass and production, including through the parameterization of tropical tree functional

types in vegetation models. This seems a promising approach to overcome the limitations imposed by the high variability of tree allometric scaling in tropical forests.



**Acknowledgements**

The BCI forest dynamics research project was founded by S.P. Hubbell and R.B. Foster and is now managed by R. Condit, S. Lao, and R. Perez under the Center for Tropical Forest Science and the Smithsonian Tropical Research in Panama. Numerous organizations have provided funding, principally the U.S. National Science Foundation, and hundreds of field workers have contributed. Data on tree morphology has been gathered through several dedicated projects and we gratefully acknowledge the contributions of P. Ramos, P. Villareal, S. Thomas, S. O'Brien, T. Spirio, and J. Dandois. IMC was supported by the Carbon Mitigation Initiative at Princeton University.

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



**Table 1.** Summary of the results of the model selection procedure. We ranked models based on Watanabe's (2013) widely Applicable Information Criterion (*WAIC*), a measure used to identify models with a good balance between predictive power (represented by the expectation of the log pointwise predictive density, *ELP*) and model complexity (represented by the estimated effective number of parameters, *pWAIC*). For each model, we report the difference in WAIC from the best model, *ΔWAIC*. We derived model weights, $w_i$, based on *WAIC* values to aid in the interpretation of the model selection procedure (Burnham, 2002). Models with the lowest *WAIC* and with *ΔWAIC* < 2 are highlighted in bold face.

| Dependent variable | Functional form[1] | Covariate[2] | *ELP* | *pWAIC* | *WAIC* | *ΔWAIC* | $w_i$ |
|---|---|---|---|---|---|---|---|
| *Tree Height* | **gMM** | **Growth** | **2725.94** | **268.69** | **-5451.89** | **0.0** | **0.92** |
| | gMM | – | 2723.14 | 266.07 | -5446.27 | 5.6 | 0.06 |
| | Weibull | Growth | 2722.06 | 270.88 | -5444.11 | 7.8 | 0.02 |
| | Weibull | – | 2720.85 | 269.53 | -5441.70 | 10.2 | 0.01 |
| | gMM | Wood density | 2717.91 | 266.57 | -5435.81 | 16.1 | 0.00 |
| | gMM | Mortality | 2716.73 | 267.23 | -5433.46 | 18.4 | 0.00 |
| | Weibull | Wood density | 2716.68 | 270.00 | -5433.36 | 18.5 | 0.00 |
| | Weibull | Mortality | 2715.88 | 269.00 | -5431.77 | 20.1 | 0.00 |
| | Power | Growth | 2178.16 | 234.16 | -4356.32 | 1095.6 | 0.00 |
| | Power | Mortality | 2175.65 | 235.69 | -4351.30 | 1100.6 | 0.00 |
| | Power | Wood density | 2174.74 | 235.68 | -4349.47 | 1102.4 | 0.00 |
| | Power | – | 2173.15 | 235.81 | -4346.31 | 1082.9 | 0.00 |
| | | | | | | | |
| *Crown area* | **Power** | **–** | **-2076.76** | **161.82** | **4153.52** | **0.0** | **0.43** |
| | **Power** | **Wood density** | **-2077.05** | **161.73** | **4154.10** | **0.6** | **0.32** |
| | Power | Mortality | -2077.80 | 163.30 | 4155.59 | 2.1 | 0.15 |
| | Power | Growth | -2078.23 | 162.76 | 4156.46 | 2.9 | 0.10 |
| | Weibull | Growth | -2097.17 | 156.29 | 4194.35 | 40.8 | 0.00 |
| | Weibull | Wood density | -2097.29 | 155.25 | 4194.58 | 41.1 | 0.00 |
| | Weibull | Mortality | -2097.88 | 154.79 | 4195.76 | 42.2 | 0.00 |
| | Weibull | – | -2099.80 | 156.82 | 4199.61 | 46.1 | 0.00 |
| | gMM | Mortality | -2118.46 | 153.38 | 4236.93 | 83.4 | 0.00 |
| | gMM | Growth | -2119.23 | 154.60 | 4238.46 | 84.9 | 0.00 |
| | gMM | – | -2120.56 | 155.88 | 4241.12 | 87.6 | 0.00 |
| | gMM | Wood density | -2120.75 | 156.56 | 4241.50 | 88.0 | 0.00 |

[1] gMM refers to the generalized Michaelis-Menten

[2] Growth refers to the log mean sapling relative growth rate, Mortality to the log mean sapling mortality rate, and Wood density to the wood specific gravity (see Methods)




**Table 2.** Posterior estimates of the parameters of the best hierarchical models for tree height and crown area allometries (see Table 1). Table entries correspond to the mean and 90% posterior central intervals for the community level parameters of each allometric function (see Eq. 4 in Methods). Tree height allometry was best described by a generalized Michaelis-Menten (gMM) model including the effect of the natural logarithm of sapling growth rate (Growth). The scaling of crown area was best described by a power law function, with similar performance between a model with no covariates and one with parameters varying depending on species wood density (Table 1). Covariates were centered and scaled before the analysis to ease comparisons of effects (natural logarithm of sapling growth rate [% yr$^{-1}$] mean (SD) = 1.01 (0.65); Wood density mean (SD) = 0.56 (0.14) g cm$^{-3}$). The standard error, $\sigma_\nu$, of the best models were 0.181 (0.179, 0.183), 0.549 (0.536, 0.563) and 0.550 (0.536, 0.562) for tree height and the two models for crown area, respectively.

|  | Functional form | Covariate | Parameter | $\alpha_\theta$ (Mean) | $\beta_\theta$ (Slope) | $\sigma_\theta$ (SD) |
|---|---|---|---|---|---|---|
| *Tree height* | gMM | Growth | $a$ | 57.0 (54.5, 60.0) | -0.093 (-0.133, -0.048) | 0.107 (0.082, 0.137) |
|  |  |  | $b$ | 0.735 (0.718, 0.752) | 0.037 (0.014, 0.060) | 0.093 (0.082, 0.105) |
|  |  |  | $k$ | 21.77 (20.70, 22.89) | -1.80 (-2.81, -0.79) | 4.18 (3.64, 4.80) |
| *Crown area* | Power | Wood density | $a$ | 0.56 (0.50, 0.63) | 0.051 (-0.001, 0.105) | 0.30 (0.26, 0.35) |
|  |  |  | $b$ | 1.35 (1.31, 1.38) | 0.011 (-0.023, 0.049) | 0.15 (0.12, 0.19) |
| *Crown area* | Power | None | $a$ | 0.57 (0.50, 0.63) | ——— | 0.30 (0.26, 0.35) |
|  |  |  | $b$ | 1.34 (1.31, 1.38) | ——— | 0.16 (0.12, 0.20) |




**Table 3.** Posterior mean estimates of total aboveground biomass density (Mg dry mass ha$^{-1}$) in the 50 ha plot in Barro Colorado Island (BCI) under alternative tree height scaling relationships. To estimate *AGB*, the height of each tree in the plot was predicted based on community- or species-level allometric models for the generalized Michaelis-Menten and power functions, together with the height-based biomass allometry equation from Chave et al. (2014) (see Methods).

|                 | *Community level* |       | *Species level* |       |
| --------------- | ----------------- | ----- | --------------- | ----- |
| *Diameter class* | Power            | gMM   | Power           | gMM   |
| 1–10 cm         | 12.6              | 12.8  | 11.9            | 12.4  |
| 10–30 cm        | 44.7              | 46.2  | 44.0            | 45.9  |
| 30–60 cm        | 78.9              | 76.0  | 80.9            | 80.4  |
| ≥60 cm          | 146.4             | 117.3 | 138.9           | 125.7 |
|                 |                   |       |                 |       |
| Total           | 282.5             | 252.2 | 275.7           | 264.4 |

Figure captions

**Figure 1.** Data (points) and best-fit allometric relationships (lines) for tree height (a) and crown area (b) in relation to trunk diameter. In each panel, blue lines correspond to species-specific fits and the white line to the community-averaged model, both from the best hierarchical model (Tables 1, 2). The best model for tree height was based on a generalized Michaelis-

Menten function (a), whereas the best model for crown area included a power function (b). Note the log scales on all axes.

**Figure 2. (a-c)** Relationships of species-specific tree height allometry parameters with log-transformed mean sapling relative growth rate, a proxy for shade-intolerance, in the best-fit hierarchical model, a model that incorporated the generalized Michaelis-Menten function (Eq. 4). Points show median posterior estimates for each individual species, with vertical bars

indicating 90% posterior central intervals. The thick grey line depicts the fitted relationship across species, and the shaded envelope encloses the 90% posterior interval. **(d)** Illustration of interspecific differences in tree height scaling in the fitted model, with the red line showing predictions for the lowest sapling growth rate (very high shade-tolerance) and the green the highest sapling growth rate (very low shade-tolerance).

**Figure 3.** Relationships of species-specific crown area allometry parameters with wood density in the second-best hierarchical model, which incorporated a power function (the best model included no covariates, Table 1). Points show median posterior estimates for each individual species, with vertical bars indicating 90% posterior central intervals. The thick grey line depicts the fitted relationship across species, while the shaded envelope encloses the 90% posterior interval.

**Figure 4.** Comparison of estimates of aboveground biomass (*AGB*, Kg dry matter) for individual trees based on measured heights (grey points) or on heights predicted from a power function fit (orange) or a generalized Michaelis-Menten fit (blue). All *AGB* estimates were based on the biomass allometry of Chave et al. (2014). Predictions from the allometric models are based on simulations of the posterior distribution (lines correspond to the median and 90% posterior central interval) of the community-level, across-species relationships.

**Figure 5.** Relative error for estimates of individual tree dry aboveground biomass (*AGB*, Kg dry matter) based on model predictions of tree height ($AGB_{\text{Hmod}}$) compared with estimates derived from height observations ($AGB_{\text{Hobs}}$). Tree height estimates were calculated using the power function (orange dots) or generalized Michaelis-Menten (blue dots) allometric models. All *AGB* estimates were based on the biomass allometry in Chave et al. (2014).

**Supplementary Material**

*Appendix S1: Extra Tables and Figures*

*Appendix S2: Stan code*





**Figure 1.** Data (points) and best-fit allometric relationships (lines) for tree height (a) and crown area (b) in relation to trunk diameter. In each panel, blue lines correspond to species-specific fits and the white line to the community-averaged model, both from the best hierarchical model (Tables 1, 2). The best model for tree height was based on a generalized Michaelis-Menten function (a), whereas the best model for crown area included a power function (b). Note the log scales on all axes.





**Figure 2.** (a-c) Relationships of species-specific tree height allometry parameters with log-transformed mean sapling relative growth rate, a proxy for shade-intolerance, in the best-fit hierarchical model, a model that incorporated the generalized Michaelis-Menten function (Eq. 4). Points show median posterior estimates for each individual species, with vertical bars indicating 90% posterior central intervals. The thick grey line depicts the fitted relationship across species, and the shaded envelope encloses the 90% posterior interval. (d) Illustration of interspecific differences in tree height scaling in the fitted model, with the red line showing predictions for the lowest sapling growth rate (very high shade-tolerance) and the green the highest sapling growth rate (very low shade-tolerance).




**Figure 3.** Relationships of species-specific crown area allometry parameters with wood density in the second-best hierarchical model, which incorporated a power function (the best model included no covariates, Table 1). Points show median posterior estimates for each individual species, with vertical bars indicating 90% posterior central intervals. The thick grey line depicts the fitted relationship across species, while the shaded envelope encloses the 90% posterior interval.





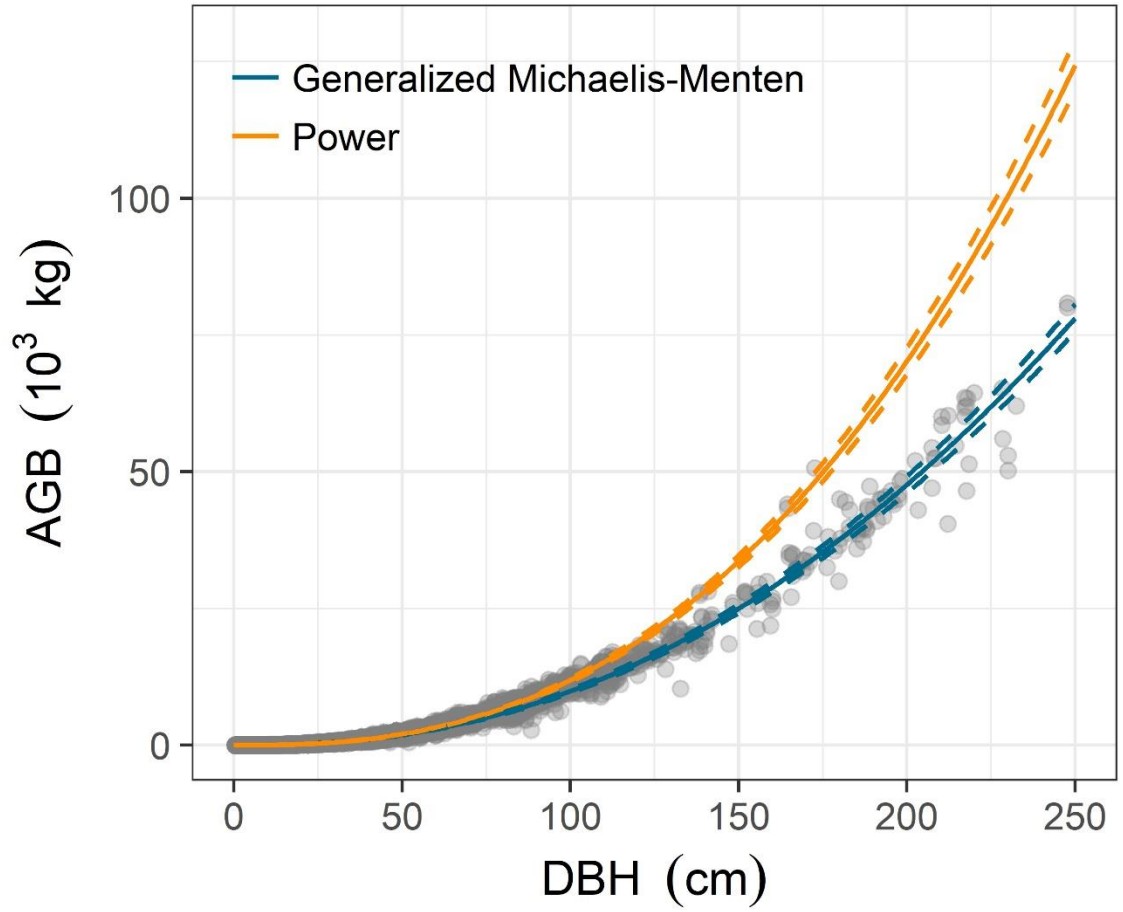

**Figure 4.** Comparison of estimates of aboveground biomass (*AGB*, Kg dry matter) for individual trees based on measured heights (grey points) or on heights predicted from a power function fit (orange) or a generalized Michaelis-Menten fit (blue). All *AGB* estimates were based on the biomass allometry of Chave et al. (2014). Predictions from the allometric models are based on simulations of the posterior
5   distribution (lines correspond to the median and 90% posterior central interval) of the community-level, across-species relationships.




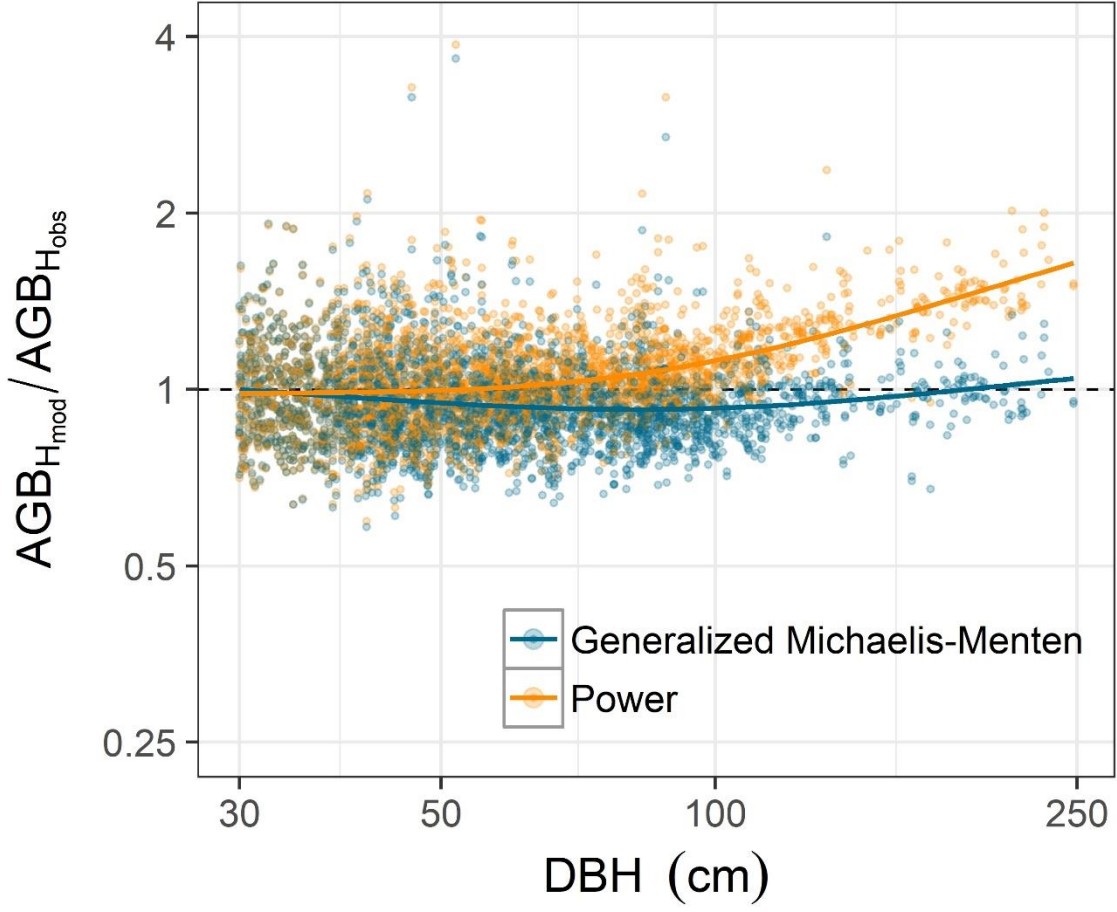

**Figure 5.** Relative error for estimates of individual tree dry aboveground biomass (*AGB*, Kg dry matter) based on model predictions of tree height (*AGB*$_{Hmod}$) compared with estimates derived from height observations (*AGB*$_{Hobs}$). Tree height estimates were calculated using the power function (orange dots) or generalized Michaelis-Menten (blue dots) allometric models. All *AGB* estimates were based on the biomass allometry in Chave et al. (2014).