# Peer review of "Interspecific variation in tropical tree height and crown allometries in relation to life history traits"

_Biogeosciences, 2018_

## Referee Comment (RC1) · N. Picard (Referee) · 12 Sep 2018

**General comments**

This study develops allometric models for tree height and crown area that integrate species traits as co-variates to account for species differences in allometric scaling. Questions (ii) and (iii) addressed by this study (P2L32-34) are not new and have been addressed by other studies before. Related text in the manuscript could be shortened (e.g. P9L10-19). Question (i) is quite new, but a recently published paper by Loubota Panzou et al. (2018) also addressed it (with a slightly different approach, though). There are differences between the results reported by Loubota Panzou et al. (2018)

[Figure]

and those shown here. For instance, Loubota Panzou et al. (2018) found that large-statured canopy species tended to be light-demanding and small-statured understory species tended to be shade-tolerant, whereas the current study found light-demanding species to have smaller stature than shade-tolerant species (Fig. 2d). Rüger et al. (2012) also reported positive (but weak) correlation between the light response of species and their maximum height. Hence, the relationship between light requirements and maximum height may be more complex than that described at P9L24-27. It has often been reported that there are many small tree species on BCI, while large light-demanding species are common in central Africa. Could it be a confounding factor?

Loubota Panzou, G.J., Ligot, G., Gourlet-Fleury, S., Doucet, J.L., Forni, E., Loumeto, J.J., and Fayolle, A. 2018. Architectural differences associated to functional traits among 45 coexisting tree species in central Africa. Functional Ecology, in press. https://doi.org/10.1111/1365-2435.13198

The implication of species differences in height allometry for forest biomass estimation is not addressed by the study. Thus the statements at P11L6 and P12L28-31 is not fully supported by the current study. Rather than comparing the AGB estimates using different height models (power model vs. saturating model), it would be more interesting to compare the gain (both in terms of accuracy and precision of the estimate of forest biomass) of including species traits into the height model. The comparison should be done at plot level and not at tree level because practical needs are for estimates of forest biomass stocks. At plot level, random tree-level errors level off, so that reducing the residual error of the allometric model at the cost of greater uncertainty on the estimates of the parameters of the model is not necessarily the best option. Uncertainty on the species traits (in particular when it comes to forest inventory data where little information is available for most species) should also be considered when assessing the relevance of integrating species traits into allometric models.

The systematic bias of AGB prediction using the power model for tree height (Figures 4 and 5) is a bit surprising. Yellow dots in Figure 5 show the ratio AGBpow/AGBobs

in log axes, where AGBpow is the biomass estimated using the power model for tree height. In other words, Fig. 5 shows log(AGBpow) − log(AGBobs). Because log(AGB) = log(0.0559) + log($\rho$) + 2log(D) + log(H), all terms except the one depending on height cancel out, so that Fig. 5 is actually showing log(Hpow) − log(Hobs), where Hpow is the height predicted by the power model. In other words, yellow dots in Fig. 5 are showing the residuals of the fitted power model for log-transformed height. If the power model for tree height had been fitted by linear regression on log-transformed data, then, by definition, these residuals would have a zero sum, which does not seem to be the case in Fig. 5 (but the $x$-axis is truncated to dbh $\geq 30$ cm, so maybe we do not have the right picture). The fact that the residuals of the power model for height (Eq. 1) are not centered on zero is a bit concerning and seems to contradict what is written at P6L16. Is it a consequence of the hierarchical approach where species-level models are averaged at community-level? Could you clarify how the community-level model is obtained from the species-level models? (Do all species have the same weight, or do all trees have the same weight? Etc.)

**Specific comments**

P2L19-20 and P10L28-29: these sentences are a bit misleading. The use of wood density as a way to account for specifies differences in multispecies biomass equation is an old idea (e.g. Brown et al. 1989) that is now commonplace. It is accordingly less common for height- and crown-diameter allometries, but see Loubota Panzou et al. (2018).

Brown, S., Gillespie, A.J.R., and Lugo, A.E. 1989. Biomass estimation methods for tropical forests with applications to forest inventory data. Forest Science 35(4): 881–902. https://doi.org/10.1093/forestscience/35.4.881

P5 Eq.(2)-(4): these expressions do not correspond to the $f$ function in Eq.(1) but rather to its exponential transform, right? In fact, the confusion comes from P4L29: Should not it be "$f$ predicts expected *log* tree height or crown area" rather than "$f$ predicts

expected tree height or crown area"? Please clarify.

P5L18: why a univariate linear function? Why not combining several traits?

P5L29: why using the same trait for all models? Why not using different traits for the different parameters of the allometric equations?

P5L29-30: the sentence is confusing. I guess "trait model" is referring to Eq.(5), but why call it a *trait* model? A model is usually called after its response variable, not after its predictors (e.g. $AGB = aD^b$ is called a biomass model, not a diameter model). Moreover, the fact that the "trait" models have twice the number of community-level parameters is not due to the fact that all models have the same predictor. It is due to the fact that each model has a single predictor.

P8L19 sqq., " (...) based on the power model": which power model? No power model to predict tree height has been presented so far (P8L2 only mentioned that power models did much worse than the other models). The power model for height is actually later given (P8L24) but without specifying on which species trait it is based (presumably the growth trait). Please clarify and present the power model before presenting the consequences for AGB estimates.

P8L24-27 and Table 3: comparing AGB estimates without specifying the estimation uncertainties does not make much sense. An AGB difference of 283 vs. 252 Mg ha$^{-1}$ does not have the same meaning if its estimation error is 10 or 1 000 Mg ha$^{-1}$. Therefore, please complement Table 3 with the estimation uncertainties. Referring to P6L14-15, please also clarifies whether you are considering only the uncertainty on the parameter estimates (confidence interval) or also the residual error (prediction interval), and possibly if you are also propagating other sources of errors (e.g. measurement error, or the error in the estimation of the species traits).

P10L7-8: unclear. What is emerging? Model fitting shows that differences among species are not strong enough for the model with a species effect to be better than

the model without a species effect. It does not show the emergence of a pattern at community level from species-level patterns. Moreover the power model for crown allometry in Farrior et al. (2016) is an assumption of the model, not an emerging property.

P10L12, P11L6 and P12L26-27: interspecific variation in crown allometry is not that "high"/"considerable"/"extensive" since the gain in predictive accuracy brought by the species trait does not even compensate for the increase in the effective number of parameters of the model.

P12L18-26: repetition of previous text.

Figure 4: what are the lines representing? I understand that you are using for height the community-level across-species relationship (Eq. 7), so that height is a function of diameter only. However the biomass equation (Eq. 6) still depends on wood density that varies across species. Therefore, one would expect to have different lines for the different species rather than a single line for all species.

Figure 5: what are the lines representing? Smoothing functions? Because Hobs is an individual tree-level data and not a one-to-one function of diameter, neither is the ratio Hmod/Hobs.

Figure 5: why are dbh starting from 30 cm instead of 1 cm?

---

## Referee Comment (RC2) · A. FAYOLLE (Referee) · 11 Oct 2018

General comments

The overall aim of the study is to examine interspecific variation in allometric scaling among coexisting species in BCI, Panama, in relation to life history traits. Allometric relationships between tree height and diameter, and between crown area and diameter, were modelled using a hierarchical Bayesian approach, allowing to identify the best functional form (saturating or not), and including trait information.

The authors identified strong interspecific variation in tree height-diameter and crowndiameter allometries, respectively related to sapling growth and wood density. They confirmed the saturating shape of the tree height-diameter relationships (best modelled with a generalized Michaelis Menten model) and showed the consequences for the estimation of biomass at the tree level, and across the 50 ha plot. Not using a saturating tree height-diameter relationship at community or species level, provided larger biomass estimates for large trees.

I really enjoyed reading the manuscript, specifically the relationships between the interspecific variation in allometric scaling and traits, though only few traits were examined... In a relatively recent work, we did find some nice relationships between crown allometry and dispersal mode among 45 coexisting species in central Africa though the inclusion of traits in the modelling was finally not included in the paper, we only kept relationships between functional traits and architectural traits derived from species-specific allometries (Loubota Panzou et al., 2018). The second aspect of the study examining consequences of height-diameter allometry on biomass estimates is more classical, and mostly confirmed previous work, though I believe it is nice to accumulate such evidence.

The way trees were sampled is not crystal clear, and additional information might be useful for the readers. In Figure 1, I would have preferred to see the raw data (not log-transformed, as in Figure 2d).

Below are some specific and sometimes really minor comments to help clarify the manuscript for the readers that might be less familiar with the study area and/or approach used.

Specific comments

P1 L11 Perhaps clarify 'finite size effects at the smallest and largest sizes' For the largest diameters, this refers to the saturation of the tree height-diameter relationships, but for the smallest diameters, it is not clear for me. Is that related to the inventory threshold?

P1 L13-14 List the trait data used since they are only 3 of them: wood density, growth and mortality. I was expected a larger set of traits at first reading.

P2 L10 The limitations of the power model for predicting the height-diameter allometry are nicely discussed in Molto et al. (2014).

P2 L23-26 I have the impression that dissociating the two arguments here, (1) the recognition of interspecific variation in allometry, and (2) the way to model it appropriately (with a hierarchical approach), would help clarify the text.

P2 L28 'a large dataset for a single site' might indicate that the 162 study species are coexisting? or do they cover multiple habitats?

P3 L15 Please consider adding a map, with the information on the old-growth and secondary forests, if possible, and sampled trees.

P3 L12 'Allometric data' sounds bizarre for me, since allometry describes relationships between tree dimensions. I would suggest 'Tree measurements' instead.

P3 L24 In the combined height-diameter and crown-diameter datasets, the number of trees differ, but the total number of species sampled remain the same (n=162 species). It would be nice to precise the average number of trees sampled per species, and the extreme values. . .

P3 L29-30 Missing space after and before '-'

P4 L1 Growth and mortality at sapling stage were considered as proxy of shade tolerance. I wonder whether more classical information on crown exposure at small size would be available on the site for the study species. The crown exposure index (CEI) at 10 or 15 cm is a good indicator of species light requirement or shade tolerance (e.g. Sheil et al., 2006), since there are always paradoxical species that deviate in their trait-performance relationships.

P5 L9-11 The generalized Michaelis Menten fitted here has 3 parameters while the 2

parameters model has shown to provide good fit as well to height-diameter allometries at plot (Molto et al., 2014) and species (Fayolle et al., 2016) levels. I did not get the advantage of including the b parameter.

P6 L3 The models were fitted after log-transformation, and this conditioned the way the results are presented (log-log scales in Figures 1 and S1). I did not get why? To be comparable with the power models?

P6 L6 The 'no trait' model includes interspecific differences, but considered random?

P7 L3 Check the units 'g cm-3'

P7 L7-8 I would have preferred to have this information on the sampling per species in the Material and Methods. . .

P8 L18 In the whole paragraph, please insist on the tree level (in kg) and forest level (50 ha pooled, in kg ha-1).

P8 L21 In the Figure 4, I do not understand the rationale behind the fit. . . AGB is predicted from Chave et al. (2014) using three predictors : wood density, diameter, and height. . . Here you have pairs of AGB estimates for each tree using height modelled with the power model and height modelled with the generalized Michaelis Menten model. . . I would suggest a simpler approach plotting the AGB_Hpow against AGB_HgMM, and the 1:1 line. . . and perhaps separately for the size classes examined in the Table 3.

P9 L8 Why 'ecological traits' here ? Please homogeneize throughout the manuscript.

P9 L19 Perhaps provide the average and range of the scaling coefficient across species for the power model, and mention the differences in estimated coefficients as well as the lack of fit of the power model.

P9 L23-26 This might be different in other tropical forests. In moist semi-deciduous forests that are widely distributed across central Africa, the forest canopy is dominated

by long-lived light demanding species... and there are only few shade tolerant species that can attain large stature... This might be different in wet forests. The analysis of the relationship between functional trait (including shade tolerance) and architectural traits describing species stature is provided in our recent paper mentioned earlier (Loubota Panzou et al., 2018).

P10 L1-24 There is a kind of contradiction between the two paragraphs: community average and interspecific variation in crown allometry.

P10 L15-18 The trait influence on crown allometry was weak. In a relatively recent work, we did find some nice relationships between crown allometry and dispersal mode among 45 coexisting species in central Africa though the inclusion of traits in the modelling was finally not included in the paper (Loubota Panzou et al., 2018).

P10 L24 Please clarify 'resource partitioning within stands', do you mean crown plasticity in response to competition?

P10 L31-32 In central Africa, using a massive destructive dataset (845 trees sampled for biomass in 6 sites), we found only little advantage of including height and crown dimensions for the prediction of AGB, possibly due to compensation between height and crown size across sites (Fayolle et al., 2018).

P11 L1-2 This has been already evidenced and discussed elsewhere (e.g. Feldpausch et al., 2011; Molto et al., 2014; Fayolle et al., 2016).

P12 L10 This has been already done elsewhere (Poorter et al., 2003, 2006; Loubota Panzou et al., 2018).

P12 L21-22 This a confirmation of previous work.

Cited references

Chave, J., Réjou-Méchain, M., Búrquez, A., Chidumayo, E., Colgan, M.S., Delitti, W.B.C., Duque, A., Eid, T., Fearnside, P.M., Goodman, R.C., Henry, M., Martínez-

Yrízar, A., Mugasha, W.A., Muller-Landau, H.C., Mencuccini, M., Nelson, B.W., Ngomanda, A., Nogueira, E.M., Ortiz-Malavassi, E., Pélissier, R., Ploton, P., Ryan, C.M., Saldarriaga, J.G., Vieilledent, G., 2014. Improved allometric models to estimate the aboveground biomass of tropical trees. Glob. Change Biol. 20, 3177–3190. https://doi.org/10.1111/gcb.12629

Fayolle, A., Loubota Panzou, G.J., Drouet, T., Swaine, M.D., Bauwens, S., Vleminckx, J., Biwole, A., Lejeune, P., Doucet, J.-L., 2016. Taller trees, denser stands and greater biomass in semi-deciduous than in evergreen lowland central African forests. For. Ecol. Manag. 374, 42–50. https://doi.org/10.1016/j.foreco.2016.04.033

Fayolle, A., Ngomanda, A., Mbasi, M., Barbier, N., Bocko, Y., Boyemba, F., Couteron, P., Fonton, N., Kamdem, N., Katembo, J., Kondaoule, H.J., Loumeto, J., Maïdou, H.M., Mankou, G., Mengui, T., Mofack, G.I., Moundounga, C., Moundounga, Q., Nguimbous, L., Nsue Nchama, N., Obiang, D., Ondo Meye Asue, F., Picard, N., Rossi, V., Senguela, Y.-P., Sonké, B., Viard, L., Yongo, O.D., Zapfack, L., Medjibe, V.P., 2018. A regional allometry for the Congo basin forests based on the largest ever destructive sampling. For. Ecol. Manag. 430, 228–240. https://doi.org/10.1016/j.foreco.2018.07.030

Feldpausch, T.R., Banin, L., Phillips, O.L., Baker, T.R., Lewis, S.L., Quesada, C.A., Affum-Baffoe, K., Arets, E., Berry, N.J., Bird, M., 2011. Height-diameter allometry of tropical forest trees. Biogeosciences 8, 1081–1106.

Loubota Panzou, G.J., Ligot, G., Gourlet-Fleury, S., Doucet, J.-L., Forni, E., Loumeto, J.-J., Fayolle, A., 2018. Architectural differences associated with functional traits among 45 coexisting tree species in Central Africa. Funct. Ecol. https://doi.org/10.1111/1365-2435.13198

Molto, Q., Hérault, B., Boreux, J.-J., Daullet, M., Rousteau, A., Rossi, V., 2014. Predicting tree heights for biomass estimates in tropical forests–a test from French Guiana. Biogeosciences 11, 3121–3130.

Poorter, L., Bongers, F., Sterck, F.J., Wöll, H., 2003. Architecture of 53 rain forest tree species differing in adult stature and shade tolerance. Ecology 84, 602–608.

Poorter, L., Bongers, L., Bongers, F., 2006. Architecture of 54 moist-forest tree species: traits, trade-offs, and functional groups. Ecology 87, 1289–1301.

Sheil, D., Salim, A., Chave, J., Vanclay, J., Hawthorne, W.D., 2006. Illumination–size relationships of 109 coexisting tropical forest tree species. J. Ecol. 94, 494–507.
* * *

---

## Referee Comment (RC3) · Anonymous Referee #3 · 18 Oct 2018

Review: bg-2018-314

Interspecific variation in tropical tree height and crown allometries in relation to life history traits

General comments

This paper uses a rather impressive dataset of nearly 10,000 observations of tree height and 2,500 observations of crown area on 162 tree species located in Barro Colorado Island, Panamá. The main results are: (i) light-demanding species attained taller heights at comparatively smaller diameters as juveniles and had shorter asymptotic heights at larger diameters as adults, and (ii) the use of saturating functional forms and

the incorporation of functional traits in tree allometric models is a promising approach to improve estimates of forest biomass and productivity. Apart from my reservations about the data collection and some mostly minor suggestions on presentation and understanding, the manuscript is very readable and the presentation is clear. However, the novelty of the present study compared to previous studies has not come across very clear to me. The first research question (How is interspecific variability in allometric scaling of tree height and crown area related to tree species functional traits, in particular wood density and measures of shade tolerance?) has been strongly studied in tropical forests, for example the study of Poorter et al. (2003, 2006) from Liberia and Bolivia, Iida et al. (2011) from Malaysia, Loubota Panzou et al. (2018) from Congo. This first research question should focus on the notion of convergence in tree allometry of coexisting tropical tree species that is not widely supported by previous studies, except for the paper Iida et al. (2011). The two last research questions have been studied in tropical forests (Fayolle et al., 2016; Ledo et al., 2016; Sullivan et al., 2018; Mensah et al., 2018). This study should focus on the effects of species or functional groups on the estimation of biomass and carbon stocks. The authors have studied the consequences of height-diameter allometry on the estimation of biomass and carbon stocks. The consequences of crown area allometry could be interesting in the estimation of biomass and carbon stocks using the allometric equation of Ploton et al. (2016).

Specific comments

The title is a bit unclear to me, because this aims to study the interspecific variation in tree allometry and its consequences on the estimation of biomass and carbon stocks.

Lines 23-24 "Our results provide an improved basis for parameterizing tropical tree functional types in vegetation models" I found "tree functional types" to be a little unclear.

I found the lines 26-27 are unclear.

Line 8: What do you mean by "tree dimensions"? A non-specialized audience (i.e. readers who are highly familiar with the literature on comparative tree biology/architecture) is likely to not specifically understand what the authors are talking about.

I found the paragraph (Lines 17-26) to be a little bit lacking in detail. First, I think the authors could include some examples of previous studies on the interspecific variation in tropical tree height and crown allometries. Secondly, the authors could include the information's on the relation between tree allometry and life-history traits. Lastly, the effects of species-specific or functional groups on the estimation of biomass should be developed in this section.

Line 8: "the vegetation is moist tropical forest". Please specify the forest type: deciduousness forest or evergreen forest?

Line 10: "... with trunk diameter of 1 cm or larger ..." What do you mean by "trunk larger"?

Line 16: I propose the "tree measurements" rather than "allometric data".

Lines 16-25: this paragraph lacks of details on the data collection. I would like the authors' give more information on the compilation of these seven datasets.

Lines 5-12: the authors may add the names of species of low height or crown area and high height or crown area.

Line 14: "... with dependence..." please reworded

Lines 16-17: "Past work suggests that mechanical resistance to self- or wind-loading cannot explain tree height allometries, as trees are generally much shorter for a given diameter than the limits based on static mechanical constraints (Niklas, 2007)". This is disconnected from your results.

Lines 12-13: Blanchard et al. (2016) study the variation inter-sites in tree allometry. Please see the reference Lines et al. (2012).

Lines 30-31: This last sentence isn't necessary.

This Fig.4 isn't necessary.

References

Fayolle, A., Loubota Panzou, G.J., Drouet, T., Swaine, M.D., Bauwens, S., Vleminckx, J., Biwole, A., Lejeune, P. & Doucet, J.-L. (2016). Taller trees, denser stands and greater biomass in semi-deciduous than in evergreen lowland central African forests. Forest Ecology and Management, 374, 42–50.

Iida, Y., Kohyama, T.S., Kubo, T., Kassim, A.R., Poorter, L., Sterck, F. & Potts, M.D. (2011). Tree architecture and life-history strategies across 200 co-occurring tropical tree species. Functional Ecology, 25, 1260–1268.

Ledo, A., Cornulier, T., Illian, J. B., Iida, Y., Kassim, A. R., & Burslem, D. F. (2016). Re‐evaluation of individual diameter: height allometric models to improve biomass estimation of tropical trees. Ecological Applications, 26(8), 2376-2382.

Lines, E. R., Zavala, M. A., Purves, D. W., & Coomes, D. A. (2012). Predictable

changes in aboveground allometry of trees along gradients of temperature, aridity and competition. Global Ecology and Biogeography, 21(10), 1017-1028.

Loubota Panzou, G. J., Ligot, G., Gourlet‐Fleury, S., Doucet, J. L., Forni, E., Loumeto, J. J., & Fayolle, A. (2018). Architectural differences associated with functional traits among 45 coexisting tree species in Central Africa. Functional Ecology.

Mensah, S., Pienaar, O. L., Kunneke, A., du Toit, B., Seydack, A., Uhl, E., ... & Seifert, T. (2018). Height–Diameter allometry in South Africa's indigenous high forests: Assessing generic models performance and function forms. Forest Ecology and Management, 410, 1-11.

Ploton, P., Barbier, N., Momo, S. T., Réjou-Méchain, M., Boyemba Bosela, F., Chuyong, G. B., ... & Henry, M. (2016). Closing a gap in tropical forest biomass estimation: taking crown mass variation into account in pantropical allometries. Biogeosciences, 13, 1571-1585.

Poorter, L., Bongers, F., Sterck, F.J. & Wöll, H. (2003). Architecture of 53 rain forest tree species differing in adult stature and shade tolerance. Ecology, 84, 511–525.

Poorter, L., Bongers, L. & Bongers, F. (2006). Architecture of 54 moist-forest tree species: traits, trade-offs, and functional groups. Ecology, 87, 1289–1301.

Sullivan, M. J., Lewis, S. L., Hubau, W., Qie, L., Baker, T. R., Banin, L. F., ... & Arets, E. (2018). Field methods for sampling tree height for tropical forest biomass estimation. Methods in Ecology and Evolution, 9(5), 1179-1189.

---

## Author Comment (AC1) · 22 Nov 2018

**Referee #1 - N. Picard**

*General comments*

This study develops allometric models for tree height and crown area that integrate species traits as co-variates to account for species differences in allometric scaling. Questions (ii) and (iii) addressed by this study (P2L32-34) are not new and have been addressed by other studies before. Related text in the manuscript could be shortened (e.g. P9L10-19). Question (i) is quite new, but a recently published paper by Loubota Panzou et al. (2018) also addressed it (with a slightly different approach, though). There are differences between the results reported by Loubota Panzou et al. (2018) and those shown here. For instance, Loubota Panzou et al. (2018) found that large-statured canopy species tended to be light-demanding and small-statured understory species tended to be shade-tolerant, whereas the current study found light-demanding species to have smaller stature than shade-tolerant species (Fig. 2d). Rüger et al. (2012) also reported positive (but weak) correlation between the light response of species and their maximum height. Hence, the relationship between light requirements and maximum height may be more complex than that described at P9L24-27. It has often been reported that there are many small tree species on BCI, while large lightdemanding species are common in central Africa. Could it be a confounding factor?

AR: We appreciate the detailed comments by the reviewer. We agree that the work by Loubota et al (2018) is relevant for the current manuscript; we did not previously cite it because it had not yet been published at the time we submitted our manuscript. We now reference Loubota et al. (2018) in multiple places. Both the approach and the results differ, as noted by the reviewer; the differences are interesting and may reflect differences in life history strategies in different forests. We agree that the relationship between light requirements and maximum height may be more complex than described in our original discussion, both in our site and in other sites, and we modified our discussion to better address these complexities. We see potential differences among regions in life history strategy distributions not as a confounding factor, but as an interesting topic for future research, although clearly beyond the current contribution or Loubota et al. 2018.

Regarding the novelty of this work, we have revised our text to more clearly emphasize the novel aspects of this study, which are to provide a sound methodological approach that allowed us to simultaneously examine species variation in tree allometry in relation to functional traits and compare alternative scaling models. We agree that other studies have addressed all three of our questions to some degree, and that some of our analyses confirm previous findings rather than presenting novel conclusions. Nonetheless, these questions continue to be of interest, as is clear from other recent publications on these topics, and from the comments of the other reviewers. We have revised the text to feature relevant prior work earlier in the introduction (the previous version treated such work mainly in the discussion.) We have also shortened some discussion text where the topics are well-treated in prior work.

The implication of species differences in height allometry for forest biomass estimation is not addressed by the study. Thus the statements at P11L6 and P12L28-31 is not fully supported by the current study. Rather than comparing the AGB estimates using different height models (power model vs. saturating model), it would be more interesting to compare the gain (both in terms of accuracy and precision of the estimate of forest biomass) of

including species traits into the height model. The comparison should be done at plot level and not at tree level because practical needs are for estimates of forest biomass stocks. At plot level, random tree-level errors level off, so that reducing the residual error of the allometric model at the cost of greater uncertainty on the estimates of the parameters of the model is not necessarily the best option. Uncertainty on the species traits (in particular when it comes to forest inventory data where little information is available for most species) should also be considered when assessing the relevance of integrating species traits into allometric models.

AR: Our manuscript did in fact address the implications of interspecific variation in height allometry for forest biomass estimates, and it reveals differences in AGB estimates for the BCI 50 ha plot based on height equations fitted for an average species with respect to those incorporating species differences in allometric parameters (Table 3, text section 3.3). We first compared individual tree AGB estimates based on actual height measurements with those obtained using allometric equations, in order to justify the use of the AGB estimates obtained using heights from the gMM model as a point of reference for the plot-level analyses (Figure 4). We then estimate plot-level biomass under alternative height models, with and without interspecific variation, and with the power or gMM models (Table 3).

The two statements highlighted by the reviewer were part of the discussion and referred to potential avenues for future research rather than specific results presented in the manuscript. We adjusted our wording to avoid any confusion. The analyses suggested by the reviewer are interesting but beyond the scope of the current contribution; at the end, this manuscript does not include any actual measurement of biomass and our allometric models did not target specifically the prediction of forest biomass. The assessment of the advantages of trait based models and the impact of uncertainty on trait measurements deserve further attention and dedicated studies on their own.

CHANGES: As detailed below, and in response to other reviewer comments, we edited the text extensively to include uncertainty measures for AGB estimates (90% credible intervals in Table 3), included figures in the text illustrating heterogeneity in species allometries, and provided further details about the comparison of AGB. As further detailed below, we now explicitly state in section 2.4.1 that our approach ignored measurement errors in trunk diameters and in species traits. Finally, we also included a new table in the supplementary material providing parameter estimates for all the models fitted; previously we included only parameter estimates for the best models (Table 2, S1).

The systematic bias of AGB prediction using the power model for tree height (Figures 4 and 5) is a bit surprising. Yellow dots in Figure 5 show the ratio AGBpow/AGBobs in log axes, where AGBpow is the biomass estimated using the power model for tree height. In other words, Fig. 5 shows log(AGBpow) − log(AGBobs). Because log(AGB) = log(0.0559) + log() + 2log(D) + log(H), all terms except the one depending on height cancel out, so that Fig. 5 is actually showing log(Hpow) − log(Hobs), where Hpow is the height predicted by the power model. In other words, yellow dots in Fig. 5 are showing the residuals of the fitted power model for log-transformed height. If the power model for tree height had been fitted by linear regression on log-transformed data, then, by definition, these residuals would have a zero sum, which does not seem to be the case in Fig. 5 (but the x-axis is truncated to dbh  30 cm, so maybe we do not have the right picture). The fact that the residuals of the power model for height (Eq. 1) are not centered on zero is a bit concerning and seems to contradict what is written at P6L16. Is it a consequence of the hierarchical approach where species-level models are averaged at community-

level? Could you clarify how the community-level model is obtained from the species-level models? (Do all species have the same weight, or do all trees have the same weight? Etc.)

AR: Figures 4 and 5 were included to highlight the better performance of the gMM model for large trees with respect to the biases found for the power function model. The reviewer is correct that the residuals are the same as those for tree height and that residuals in log(AGB) sum to zero across the fitted data when the full range of DBHs and random effects are considered. However, plot-level estimates of AGB are concerned with the sum of residuals in AGB (not log(AGB)) over trees with the observed plot-level DBH distribution (not the DBH distribution of the trees included in the allometric model fitting), and these do not sum to zero since the contribution of large trees outweigh the underestimates in many small trees. Our aim with Figure 5 was to illustrate this bias in large trees, so we plotted a small subset of DBH ranges to avoid the clutter at very small sizes. We have now included another version of this figure featuring the full range of DBH in the supplementary materials (Fig S3).

It is important to note that the AGB predictions in figures 4 and 5 do not include species-specific information; they are based on the community level model and use an average wood density. The community level height model features expected parameter values for an average species. The hierarchical structure accounts for potential differences in sampled DBH ranges and in sample size across species. The bias in the prediction of the power function reflects the inability of this functional form to accommodate the saturation and convergence in tree height at large diameter sizes; indeed, the misfit at larger sizes is compensated by deviations at small tree sizes. The statement in P6L16 about the unbiased community-level models refer in the case of tree height to the gMM model, which provides a better fit and much better residual patterns than the power function (at the end, we rejected power function models for tree height).

CHANGES: As detailed below, we revised the presentation of the AGB comparisons in the methods and in the results. We also revised the captions of figures 4 and 5 and combined both plots in a single figure. We included a new figure in the supplementary material featuring the full range of DBHs for the comparison of AGB.

*Specific comments*

P2L19-20 and P10L28-29: these sentences are a bit misleading. The use of wood density as a way to account for specifies differences in multispecies biomass equation is an old idea (e.g. Brown et al. 1989) that is now commonplace. It is accordingly less common for height- and crown-diameter allometries, but see Loubota Panzou et al. (2018).

AR: Good point; we edited the text to account for the common usage of wood density. In the first case, we were referring to interspecific differences in allometric scaling for different tree dimensions, so we edited the text to make clear the distinction.

CHANGE: [P2L19] *Approaches pooling data across species inherently fail to recognize species heterogeneity in allometric scaling and limit the potential to identify and define plant functional groups.*

[P10L28] *Whereas many models incorporate only individual trunk diameter (Brown, 1997) and species wood density (Brown et al. 1989), current state-of-the-art models typically include estimates of  tree height as well (e.g., Chave et al., 2014), and crown dimensions have also been incorporated in some models (Goodman et al., 2014; Ploton et al., 2016; although Fayolle et al. 2018 found a minor role of either crown or height dimensions on biomass estimates).*

P5 Eq.(2)-(4): these expressions do not correspond to the f function in Eq.(1) but rather to its exponential transform, right? In fact, the confusion comes from P4L29: Should not it be "f predicts expected log tree height or crown area" rather than "f predicts expected tree height or crown area"? Please clarify.

AR: Good point; we corrected the text as suggested.

CHANGE: [eq (1)] *where the process model, f, predicts expected natural log tree height or crown area* […]

P5L18: why a univariate linear function? Why not combining several traits?

P5L29: why using the same trait for all models? Why not using different traits for the different parameters of the allometric equations?

AR: We agree that it would in principle be interesting to evaluate additional different, more complex models, incorporating multiple traits, more alternative traits, more complex relationships with traits, etc. However, even our large dataset has power limitations. Fitting more parameter-rich models invariably results in greater uncertainty in individual parameter estimates, and a data mining approach of fitting large numbers of alternative models has known disadvantages with respect to interpretability and reproducibility. Our work fits more complex models than most previous studies, compares a limited number of models involving traits chosen based on previous work and first principles, and incorporates these traits in the simplest possible framework (univariate, linear, one trait at a time). Future research based on larger datasets could usefully evaluate more different and more complex models.

P5L29-30: the sentence is confusing. I guess "trait model" is referring to Eq.(5), but why call it a trait model? A model is usually called after its response variable, not after its predictors (e.g. AGB = aDb is called a biomass model, not a diameter model). Moreover, the fact that the "trait" models have twice the number of community-level parameters is not due to the fact that all models have the same predictor. It is due to the fact that each model has a single predictor.

AR: We are comparing multiple models for, e.g., height, and we need to distinguish them, so it would be confusing to refer to them all as "height models". We could refer to them as "model 1" and "model 2", etc., but preferred instead an informative name that provides information on what distinguishes the models, which in our case is in part whether they are based on traits or not. We agree that any model in which each parameter had a single predictor would have the same number of parameters, and we did not mean our wording to suggest otherwise.

CHANGE: We revised the text to avoid any confusion regarding the use of the "trait model" naming convention. Where necessary, we detailed whether we were referring to height or crown area models featuring the effect of a trait. With respect to the trait models having twice the number of parameters, we clarify the text as follows;

*We refer to models incorporating relationships between allometric parameters and species traits as 'trait models'. Because each allometric parameter was a linear function of a trait,  trait models had twice  as many community-level parameters as corresponding models lacking covariates. Our trait models each featured a single trait (all parameters in a trait model depended on the same trait). *

P8L19 sqq., " (...) based on the power model": which power model? No power model to predict tree height has been presented so far (P8L2 only mentioned that power models did much worse than the other models). The power model for height is actually later given (P8L24) but without specifying on which species trait it is based (presumably the growth trait). Please clarify and present the power model before presenting the consequences for AGB estimates.

AR: We thank the reviewer for drawing our attention to this oversight. We now introduce the equation for the tree height, community level power model used in Figures 4 and 5 earlier in the text. We also edited the text to detail which type of model was used for each comparison. Figures 4 and 5 are based on community level, height-diameter models with traits, although the inclusion of traits does not affect the divergence in AGB predictions. Table 3 details the impact of taking into account species identity using the 50 ha BCI plot.

CHANGE: [First sentence Section 3.3] *Individual tree aboveground biomass (AGB) estimates based on the community-average power model ($H = 3.02D^{0.56}$, Table S2) were strongly upwardly biased for large trees relative to estimates based on measured heights, whereas AGB estimates based on the gMM height model were unbiased (Fig. 4).*

P8L24-27 and Table 3: comparing AGB estimates without specifying the estimation uncertainties does not make much sense. An AGB difference of 283 vs. 252 Mg ha−1 does not have the same meaning if its estimation error is 10 or 1 000 Mg ha−1. Therefore, please complement Table 3 with the estimation uncertainties. Referring to P6L14-15, please also clarifies whether you are considering only the uncertainty on the parameter estimates (confidence

interval) or also the residual error (prediction interval), and possibly if you are also propagating other sources of errors (e.g. measurement error, or the error in the estimation of the species traits).

AR: We added uncertainty estimates for plot-level AGB using 90% credible intervals based on 5000 samples from the posterior distributions of all the parameters of our allometric models (paralleling those given for other parameters and derived quantities features in the manuscript. This addition does not alter the interpretation or conclusions of the analysis of differences in AGB estimates across models. We ignored measurement errors in trunk diameters and in species traits, as we now explicitly state in section 2.4.1. We considered these sources of error to be minor compared to measurement errors in height and crown area, although we agree their full consideration would be a useful addition to future work. We also noted to readers that we ignored uncertainty in the Chave et al (2014) AGB model.

CHANGE: [End of Section 2.4.1] *Finally, the model ignored measurement errors in trunk diameter and in ancillary trait data.*

[Section 2.4.3]: *We computed 90% credible intervals for each AGB estimate based on 5000 samples from the posterior distributions of all parameters of the corresponding allometric models.*

[Table 3]: *Posterior mean estimates (with their 90% credible intervals) of total aboveground biomass*

P10L7-8: unclear. What is emerging? Model fitting shows that differences among species are not strong enough for the model with a species effect to be better than the model without a species effect. It does not show the emergence of a pattern at community level from species-level patterns. Moreover the power model for crown allometry in Farrior et al. (2016) is an assumption of the model, not an emerging property.

P10L12, P11L6 and P12L26-27: interspecific variation in crown allometry is not that "high"/"considerable"/"extensive" since the gain in predictive accuracy brought by the species trait does not even compensate for the increase in the effective number of parameters of the model.

AR: We agree that the wording regarding "emergence" was unclear, and have revised it.  The models with no traits also feature variation in allometric parameters among species, it is just that this variation is not linked to the traits included in the model. Individual effects differed markedly among species both for tree height and crown area (see Figs 2 and 3). The estimated crown area exponent b ranges from 1.09 to 1.77 across species, which is accurately described as "high", "considerable" or "extensive" variation. Following a comment by reviewer R#2, we provide a summary of the information available in Table S2 in section 4.2 to support the statements about the variability in crown area scaling across species.

CHANGE: *This large-scale consistency in community-level relationships emerges despite local variation among species (e.g. the exponent b ranged between 1.09 and 1.77 across species, Table S2).*

AR: We tried to summarize the main points of the manuscript in the last, concluding paragraph. We believe that a certain degree of repetition is allowed (and recommended) in this context.

Figure 4: what are the lines representing? I understand that you are using for height the community-level across-species relationship (Eq. 7), so that height is a function of diameter only. However the biomass equation (Eq. 6) still depends on wood density that varies across species. Therefore, one would expect to have different lines for the different species rather than a single line for all species.

AR: The lines in the figure portray predictions based on community level allometric equations and community average wood density, for comparison with predictions using individual height and community average wood density (points). We revised the relevant text in the methods and the figure caption to clarify these details. We use the same wood density for all species to highlight the effects of the height allometry; as pointed by the reviewer, we note that the proportional error for each tree would be the same if we included species-specific wood densities in both cases instead. AGB predictions based on models taking into account species differences in wood density and allometric scaling are compared in Table 3. As noted in the text, taking into account species identity can improve the quality of AGB estimates based on a power function model for tree height, although they do not correct for biases at large DBHs.

CHANGE: [Section 2.4.3] *For those species for which species-specific wood densities were not available, we used the average over species for which values were available ($\rho$ = 0.5304 g cm$^{-3}$, Wright et al., 2010).*

[Figure 4 caption] *Comparison of estimates of aboveground biomass (AGB, Kg dry matter) as a function of DBH based on observed tree heights (grey points) with those based on height predicted from community level power function (orange lines) or generalized Michaelis-Menten (blue lines) models. for individual trees based either on measured heights (grey points) or on heights predicted from a power function fit (orange) or a generalized Michaelis-Menten fit (blue). All AGB estimates were based on the biomass allometry of Chave et al. (2014) and used the average value of wood density across species ($\rho$ = 0.5304 g cm$^{-3}$; data from Wright et al., 2010) to highlight variation related to the height allometry. Predictions from the allometric models are based on simulations of the posterior distribution (lines correspond to the median and 90% posterior central interval) of the community-level, across-species relationships.*

Figure 5: what are the lines representing? Smoothing functions? Because Hobs is an individual tree-level data and not a one-to-one function of diameter, neither is the ratio Hmod/Hobs.

Figure 5: why are dbh starting from 30 cm instead of 1 cm?

AR: Good point; we edited the figure caption to detail both aspects. The lines are LOESS smoothers and they were added to show the overall departure of each model from observations for different diameter ranges. We restricted the range to focus on the large tree sizes where the predictions of the power and gMM model differed systematically, and which drive differences in plot-level estimates. We have now added a figure showing the full range of diameters to the supplementary materials.

CHANGE: [Figure 5 caption] *Relative error for estimates of individual tree dry aboveground biomass (AGB, Kg dry matter) based on model predictions of tree height (AGBHmod) compared with estimates derived from height observations (AGBHobs), for trees with DBH > 30 cm (the full range is shown in figure S3). Modeled tree heights were from community-level models fitted with either the power function (orange dots) or generalized Michaelis-Menten function (blue dots). All AGB estimates were based on the biomass allometry equation 6 (from Chave et al. 2014) and used the average value of wood density across species, to highlight variation related to the height allometry. The lines are LOESS smoothers that illustrate the overall departures of each model from perfect prediction (i.e. AGBHmod/AGBHobs ratio equal to unity) as a function of DBH.*

We added a new figure S3 in the supplemental material featuring deviations over the full range of DBHs.

---

## Author Comment (AC2) · 22 Nov 2018

**Referee #2 - A. FAYOLLE**

*General comments*

The overall aim of the study is to examine interspecific variation in allometric scaling among coexisting species in BCI, Panama, in relation to life history traits. Allometric relationships between tree height and diameter, and between crown area and diameter, were modelled using a hierarchical Bayesian approach, allowing to identify the best functional form (saturating or not), and including trait information.

The authors identified strong interspecific variation in tree height-diameter and crown- diameter allometries, respectively related to sapling growth and wood density. They confirmed the saturating shape of the tree height-diameter relationships (best modelled with a generalized Michaelis Menten model) and showed the consequences for the estimation of biomass at the tree level, and across the 50 ha plot. Not using a saturating tree height-diameter relationship at community or species level, provided larger biomass estimates for large trees.

I really enjoyed reading the manuscript, specifically the relationships between the interspecific variation in allometric scaling and traits, though only few traits were examined. . . In a relatively recent work, we did find some nice relationships between crown allometry and dispersal mode among 45 coexisting species in central Africa though the inclusion of traits in the modelling was finally not included in the paper, we only kept relationships between functional traits and architectural traits derived from species-specific allometries (Loubota Panzou et al., 2018). The second aspect of the study examining consequences of height-diameter allometry on biomass estimates is more classical, and mostly confirmed previous work, though I believe it is nice to accumulate such evidence.

The way trees were sampled is not crystal clear, and additional information might be useful for the readers. In Figure 1, I would have preferred to see the raw data (not log-transformed, as in Figure 2d).

Below are some specific and sometimes really minor comments to help clarify the manuscript for the readers that might be less familiar with the study area and/or approach used.

> AR: We really appreciate the feedback including the detailed and useful suggestions. We took advantage of them and those by the other reviewers to improve the manuscript and amend all the weak points highlighted. We included the references suggested, framing and discussing our results in the context of previous work. We also emphasized the novel aspects of this study, with a sound methodological approach that allowed us simultaneously to examine species variation in tree allometry in relation to functional traits and to compare alternative models. We provided more details on the data collection following your recommendations and explained the rationale behind fitting the model in log scale.

*Specific comments*

P1 L11 Perhaps clarify 'finite size effects at the smallest and largest sizes' For the largest diameters, this refers to the saturation of the tree height-diameter relationships, but for the smallest diameters, it is not clear for me. Is that related to the inventory threshold?

AR: We revised the text and avoided the term 'finite size effects' (e.g. Enquist & Bentley 2012 *in* Sibly et al. *Metabolic Ecology*). Allometric exponents generally decrease with individual tree size, and these size-associated changes have been associated with changes in architecture and wood density during early development, and with the proportionally larger investment in reproduction of larger trees.

CHANGE: *In tropical forests, allometric relationships are often modeled by fitting scale-invariant power functions to pooled data from multiple species, an approach that fails to capture changes in scaling during ontogeny and physical limits to maximum tree size, and that ignores interspecific differences in allometry.*

P1 L13-14 List the trait data used since they are only 3 of them: wood density, growth and mortality. I was expected a larger set of traits at first reading.

CHANGE: [...] *Here, we analyzed allometric relationships of tree height (9884 individuals) and crown area (2425) with trunk diameter for 162 species from Barro Colorado Island, Panamá using species-specific data on wood density and sapling growth and mortality.*

P2 L10 The limitations of the power model for predicting the height-diameter allometry are nicely discussed in Molto et al. (2014).

AR: Good point, we added an additional citation to Molto et al. (2014).

P2 L23-26 I have the impression that dissociating the two arguments here, (1) the recognition of interspecific variation in allometry, and (2) the way to model it appropriately (with a hierarchical approach), would help clarify the text.

AR: We restructured the text as suggested.

CHANGE: *Species differ systematically in allometric relationships, suggesting that these differences reflect underlying interspecific variation in life-history, physiology, morphology, and/or phylogeny (Westoby et al., 2002; Adler et al., 2014). Hierarchical approaches based on functional traits can provide a useful approach for capturing this interspecific variation in tree allometry (Dietze et al., 2008; Iida et al., 2011).*

P2 L28 'a large dataset for a single site' might indicate that the 162 study species are coexisting? or do they cover multiple habitats?

AR: All the crown area data were collected in one 50 ha plot, and we only included species for which we had crown area data. So, all species co-occur within an area of 50 ha. Some species are widespread within this area, and some are associated with different habitats defined by topography or canopy height (Harms et al. 2001; Dalling et al. 2012). Whether they all stably coexist within this area rather than simply co-occur is debated. Approximately half the tree height data were collected on a 38.4 ha plot on Gigante, ~6 km away, on species also found on the 50 ha plot. We don't think this can be considered a multi-site study, so it seems appropriate to refer to it as a single site.

P3 L15 Please consider adding a map, with the information on the old-growth and secondary forests, if possible, and sampled trees.

CHANGE: We referred readers to Mascaro et al. 2011 who provide a detailed description of BCI, including a map with the distribution of old-growth and secondary forests in the island. For the Gigante peninsula, we referred readers to Wright et al. 2011, who provides a map, and to Denslow and Guzman 2000, who give forest age for selected plots.

P3 L12 'Allometric data' sounds bizarre for me, since allometry describes relationships between tree dimensions. I would suggest 'Tree measurements' instead.

CHANGE: Subsection title revised to: *2.2 Tree measurements.*

P3 L24 In the combined height-diameter and crown-diameter datasets, the number of trees differ, but the total number of species sampled remain the same (n=162 species). It would be nice to precise the average number of trees sampled per species, and the extreme values. . .

CHANGE: We included the requested information on each panel of Fig S1 and alerted readers of the availability of this information in the main text.

P3 L29-30 Missing space after and before '-'

AR: Corrected, thanks.

P4 L1 Growth and mortality at sapling stage were considered as proxy of shade tolerance. I wonder whether more classical information on crown exposure at small size would be available on the site for the study species. The crown exposure index (CEI) at 10 or 15 cm is a good indicator of species light requirement or shade tolerance (e.g. Sheil et al., 2006), since there are always paradoxical species that deviate in their trait-performance relationships.

AR: We agree that in principle this could be an even better proxy for shade-tolerance, but it is not available for our species. Fortunately, previous research shows that sapling growth and mortality rates are excellent proxies for shade-tolerance (Wright et al. 2010).

P5 L9-11 The generalized Michaelis Menten fitted here has 3 parameters while the 2 parameters model has shown to provide good fit as well to height-diameter allometries at plot (Molto et al., 2014) and species (Fayolle et al., 2016) levels. I did not get the advantage of including the b parameter.

AR: The third parameter essentially adds flexibility to the traditional Michaelis-Menten model, which becomes a special case corresponding to the exponent b =1. When the exponent b is less than one, the log-log slope of the height-diameter relationship at small sizes is shallower than in the traditional Michaelis-Menten model. The best generalized Michaelis-Menten model for height had a cross-species mean exponent b = 0.73, substantially and significantly different from 1 (90% CI 0.72, 0.75), and the vast majority of species-specific exponents were also significantly different from 1 (Table S2). Thus, it is clear that the generalized Michaelis-Menten is a much better fit than a traditional Michaelis-Menten with b=1. We note that this issue was addressed in the discussion, in section 3.1.

CHANGE: We added the suggested references.

P6 L3 The models were fitted after log-transformation, and this conditioned the way the results are presented (log-log scales in Figures 1 and S1). I did not get why? To be comparable with the power models?

AR: The residuals of height and crown area are heteroscedastic, with increasing variance at increasing diameters.  Log-transformation makes the residuals essentially homoscedastic, simplifying model fitting. In general, allometric data tend to show this type of proportional variation, and are thus well-suited to analysis after log-transformation.

CHANGE: We included a citation to Mascaro et al 2014, a manuscript discussing log transformation in allometric studies.

P6 L6 The 'no trait' model includes interspecific differences, but considered random?

AR: Yes, as detailed in the description of the models in P5L27, 'no trait' models regard variation in allometric parameters among species as random. We revised the text to remind readers about that;

CHANGE: [...] *four possibilities for functional traits (wood density, sapling growth, and sapling mortality, but also 'no trait' models featuring only random variability in allometric parameters across species).*

P7 L3 Check the units 'g cm-3'

CHANGE: We corrected the typo.

P7 L7-8 I would have preferred to have this information on the sampling per species in the Material and Methods

CHANGE: We moved this sentence to the Materials and Methods (section 2.2).

P8 L18 In the whole paragraph, please insist on the tree level (in kg) and forest level (50 ha pooled, in kg ha-1).

CHANGE: We revised the text to detail the units of each biomass quantity and to prevent any confusion; thanks for the suggestion.

P8 L21 In the Figure 4, I do not understand the rationale behind the fit. . . AGB is predicted from Chave et al. (2014) using three predictors : wood density, diameter, and height. . . Here you have pairs of AGB estimates for each tree using height modelled with the power model and height modelled with the generalized Michaelis Menten model. . . I would suggest a simpler approach plotting the AGB_Hpow against AGB_HgMM, and the 1:1 line. . . and perhaps separately for the size classes examined in the Table 3.

AR: We have stressed in the text and in the caption of Figure 4 that there is no fit since there are no AGB measurements. We choose to graph the data in this way because we wanted to highlight the increasing divergence at larger diameters.  A 1:1 graph would show the relative difference in AGB at larger AGB, but would provide no link to diameters.

CHANGES: Section 3.3 – *AGB estimates calculated using tree height predictions based on the power model*  *exceeded those based on the gMM model*  *by ever larger proportions at higher trunk diameters, with an overestimate of 10% at D = 66 cm [52, 80]90%, reaching 59% [51, 67]90% at D = 250 cm (Figs. 4 and 5).*

[Figure 4 caption] *Comparison of estimates of aboveground biomass (AGB, Kg dry matter) as a function of DBH based on observed tree heights (grey points) with those based on height predicted from community level power function (orange lines) or generalized Michaelis-Menten (blue lines) models.*  *All AGB estimates were based on the biomass allometry of Chave et al. (2014) and used the average value of wood density across species ($\rho$ = 0.5304 g cm$^{-3}$; data from Wright et al., 2010) to highlight variation related to the height allometry. Predictions from the allometric models are based on simulations of the posterior distribution (lines correspond to the median and 90% posterior central interval) of the community-level, across-species relationships.*

P9 L8 Why 'ecological traits' here ? Please homogeneize throughout the manuscript.

AR: Good point; we now use "functional traits" throughout the text.

P9 L19 Perhaps provide the average and range of the scaling coefficient across species for the power model, and mention the differences in estimated coefficients as well as the lack of fit of the power model.

AR: We revised the text at this location to reference the mismatch in exponents and the lack of fit of the power model. Given the poor fit of the power model in general, we do not think it is useful to discuss the fitted values in more detail.

CHANGE: *Metabolic theories based on hydraulic constraints predict a constant logarithmic scaling between tree height and trunk diameter with an exponent close to 2/3 (Niklas and Spatz, 2004; West et al., 2009), inconsistent with our results, which show that community-level power function exponents differ significantly from 2/3, and that the data diverge strongly from the power function.*

P9 L23-26 This might be different in other tropical forests. In moist semi-deciduous forests that are widely distributed across central Africa, the forest canopy is dominated by long-lived light demanding species. . . and there are only few shade tolerant species that can attain large stature. . . This might be different in wet forests. The analysis of the relationship between functional trait (including shade tolerance) and architectural traits describing species stature is provided in our recent paper mentioned earlier (Loubota Panzou et al., 2018).

14

AR: We agree that our original wording gave the misleading impression that there was a strong and consistent relationship between shade-tolerance and maximum stature among tree species within tropical forests.  We have revised our wording to make clear that this is not the case, not even in our own focal site.

CHANGE: *Interspecific variation in tree height scaling parameters was associated with sapling growth rates, which suggests a tendency for shade-tolerance and allometric strategies to be aligned in this community (Wright et al. 2010).  At one extreme are fast-growing, light-demanding tree species that have larger heights at small stem diameters; at the other extreme, slow-growing, shade-tolerant species have higher heights at larger diameters (Bohlman and O'Brien, 2006), and higher asymptotic heights (parameter a). This does not mean that shade-tolerant species tend to have larger maximum heights, because maximum heights depend on maximum diameters and are often much less than asymptotic heights for small-statured species (Figure S1). In general, shade-tolerance and maximum height are largely independent axis of variation among tropical tree species (Bohlman and Pacala 2012; Rüger et al., 2018), and may if anything tend to be negatively correlated across species (Poorter et al. 2006, Loubota-Panzou et al. 2018, Wright et al. 2010). The differences in allometric parameters should be interpreted in terms of differences in trajectories, especially at small diameters, where light-demanding species take greater risks.*

P10 L1-24 There is a kind of contradiction between the two paragraphs: community average and interspecific variation in crown allometry.

AR: Indeed, we tried to highlight the contrast between the consistency in community average allometries across sites, and the variation among species within sites, which we agree may at first seem contradictory. We now detail the range of exponents estimated across species (b = [1.09, 1.77]) to clarify the latter point. We also revised the wording of the first sentence (which previously began "Crown area presented a constant scaling with trunk diameter") to avoid confusion regarding what is meant by "constant".

CHANGE: *Crown area and trunk diameter presented a scale-invariant relationship, with no indication of saturation even for the largest trees in our dataset.  [...]  This large-scale consistency in community-level relationships emerges despite local variation among species (e.g. the exponent b ranged between 1.09 and 1.77 among our species, Table S2). Modeling studies show that community-level crown area allometric parameters crucially determine the scaling of tree growth and mortality and the parameters of tree size distributions (Muller-Landau et al. 2006 a,b; Farrior et al., 2016).*,

P10 L15-18 The trait influence on crown allometry was weak. In a relatively recent work, we did find some nice relationships between crown allometry and dispersal mode among 45 coexisting species in central Africa though the inclusion of traits in the modelling was finally not included in the paper (Loubota Panzou et al., 2018).

AR: We added a citation to Loubota Panzou et al. (2018) highlighting that maximum tree height and maximum crown area tend to be larger in wind dispersed species.

CHANGE: *However, other traits might explain these differences. For instance, Louboza Panzou et al. (2018) found that wind-dispersed species had taller heights and larger crown dimension.*

P10 L24 Please clarify 'resource partitioning within stands', do you mean crown plasticity in response to competition?

AR: We revised the wording as suggested:

CHANGE: [...] *other factors can be important in shaping crown geometry in large trees, including crown plasticity in response to competition (Thomas, 1996; Poorter et al. 2008).*

P10 L31-32 In central Africa, using a massive destructive dataset (845 trees sampled for biomass in 6 sites), we found only little advantage of including height and crown dimensions for the prediction of AGB, possibly due to compensation between height and crown size across sites (Fayolle et al., 2018).

AR: Thank you for pointing us to this impressive study, which was published while this manuscript was under review. We added a citation to this publication following references to Goodman et al (2014) and Ploton et al (2016) to note the contrasting results.

CHANGE: [P10L30] [...] *crown dimensions have also been incorporated in some models (Goodman et al., 2014; Ploton et al., 2016; although Fayolle et al. 2018 found a minor role of either crown or height dimensions on biomass estimates)*

[P12L3] *Finally, we evaluated only crown area, even though crown depth and crown shape are also important for the estimation of tree biomass (Goodman et al., 2014; Ploton et al., 2016; but see also Fayolle et al. 2018) and for characterizing tree life history strategies (Canham et al., 1994; Poorter et al. 1996; Bohlman and O'Brien, 2006).*

P11 L1-2 This has been already evidenced and discussed elsewhere (e.g. Feldpausch et al., 2011; Molto et al., 2014; Fayolle et al., 2016).

AR: True, and we did not intend to suggest that our results were novel in this respect.

CHANGE: We have reworded, changing "*highlight*" to "*confirm*", and added the references at this point to make this clear.

P12 L10 This has been already done elsewhere (Poorter et al., 2003, 2006; Loubota Panzou et al., 2018).

AR: We have revised the text to include the above-mentioned references.

P12 L21-22 This a confirmation of previous work.

CHANGE: We changed "*show*" to "*confirm*" and added citations.

*References*

Mascaro, J., G. P. Asner, H. C. Muller-Landau, M. Van Breugel, J. Hall, and K. Dahlin. 2011. Controls over aboveground forest carbon density on Barro Colorado Island, Panama. Biogeosciences 8:1615-1629.

Denslow, J. S., and Guzman G., S. 2000. Variation in stand structure, light and seedling abundance across a tropical moist forest chronosequence, Panama, J. Veg. Sci., 11, 201-212.

Wright, S. J., J. B. Yavitt, N. Wurzburger, B. L. Turner, E. V. J. Tanner, E. J. Sayer, L. S. Santiago, M. Kaspari, L. O. Hedin, K. E. Harms, M. N. Garcia, and M. D. Corre. 2011. Potassium, phosphorus, or nitrogen limit root allocation, tree growth, or litter production in a lowland tropical forest. Ecology 92:1616-1625.

---

## Author Comment (AC3) · 22 Nov 2018

Referee #3

*General comments*

This paper uses a rather impressive dataset of nearly 10,000 observations of tree height and 2,500 observations of crown area on 162 tree species located in Barro Colorado Island, Panamá. The main results are: (i) light-demanding species attained taller heights at comparatively smaller diameters as juveniles and had shorter asymptotic heights at larger diameters as adults, and (ii) the use of saturating functional forms and the incorporation of functional traits in tree allometric models is a promising approach to improve estimates of forest biomass and productivity. Apart from my reservations about the data collection and some mostly minor suggestions on presentation and understanding, the manuscript is very readable and the presentation is clear. However, the novelty of the present study compared to previous studies has not come across very clear to me.

> AR: We appreciate the feedback provided and took advantage of the suggestions raised by the reviewer to prepare an improved version of the manuscript. We included all the new references suggested, framing and discussing our results in the context of previous work. We have also emphasized the novel aspects of this study, which provides a sound methodological approach that enabled us to simultaneously examine species variation in tree allometry in relation to functional traits and to compare alternative scaling models.

The first research question (How is interspecific variability in allometric scaling of tree height and crown area related to tree species functional traits, in particular wood density and measures of shade tolerance?) has been strongly studied in tropical forests, for example the study of Poorter et al. (2003, 2006) from Liberia and Bolivia, Iida et al. (2011) from Malaysia, Loubota Panzou et al. (2018) from Congo. This first research question should focus on the notion of convergence in tree allometry of coexisting tropical tree species that is not widely supported by previous studies, except for the paper Iida et al. (2011).

> AR: We revised the introduction to better frame the questions pursued in the manuscript. We now reference the work by Poorter et al (2006) earlier in the manuscript and indicate that they pursued similar questions (as di Ilda et al. 2011). We also revised the reference list to add the work by Loubota et al. (2018), which was published after this manuscript was submitted. This recent publication in a leading journal demonstrates that the first research question remains of interest, and we kept this question in its current form. We agree that convergence (or lack thereof) in tree allometries is an interesting topic, and we address this in the discussion section, as in the previous version. In our view, this question can be best addressed through analyses that encompass multiple sites, and is thus beyond the scope of this study).

The two last research questions have been studied in tropical forests (Fayolle et al., 2016; Ledo et al., 2016; Sullivan et al., 2018; Mensah et al., 2018). This study should focus on the effects of species or functional groups on the estimation of biomass and carbon stocks. The authors have studied the consequences of height-diameter allometry on the estimation of biomass and carbon stocks. The consequences of crown area allometry could be interesting in the estimation of biomass and carbon stocks using the allometric equation of Ploton et al. (2016).

> AR: We revised the introduction to include the suggested citations, some of which were already mentioned later in the manuscript (Fayolle et al 2016; Ledo et al. 2016, Sullivan et al. 2018). The recent

publication of these studies demonstrates that there is still wide interest in the questions pursued in this study. We believe that our methods and results contribute usefully to the ongoing debate about which allometric model should be employed to describe tree height and crown area. We hope that the addition of the suggested references and the minor edits clarify this point. We agree that it would be interesting to address the consequences of crown area allometries for biomass and carbon stocks, but we preferred to stick with our simpler approach of only addressing the consequences of height allometries. There is a huge body of literature dealing with AGB estimation, with a corresponding variety of methods and equations. The models by Chave et al (2014) that require only diameter, height and wood density are based on the largest database of AGB measurements available (an order of magnitude above the number of trees measured to develop Ploton et al. 2016 models), and are some of the most widely used. The aim of the comparison presented in Fig 4 and in Table 3 was to highlight how height saturation and species variation interact to affect AGB estimation. The use of a model including crown area would not alter our conclusions. In our opinion, it might even distract readers from the main message; in the end, our data set analyzed does not include biomass measurements and thus cannot clarify which model is truly better.

*Specific comments*

The title is a bit unclear to me, because this aims to study the interspecific variation in tree allometry and its consequences on the estimation of biomass and carbon stocks.

AR: As commented above, the analyses of biomass were just included to highlight the impact of model choice in tree allometric studies. However, the main subject of the manuscript lies is the adequacy of alternative allometric models and the relationship of allometric parameters to ecological traits across species.

Lines 23-24 "Our results provide an improved basis for parameterizing tropical tree functional types in vegetation models" I found "tree functional types" to be a little unclear.

CHANGE: We revised the term to "Plant functional types (PFTs)" and included a citation to (Prentice et al., 1992)

I found the lines 26-27 are unclear.

CHANGE: *Allometric scaling  describes how evolutionary and physical constraints on plant morphology and performance vary as a function of size  (Niklas, 1994).*

Line 8: What do you mean by "tree dimensions"? A non-specialized audience (i.e. readers who are highly familiar with the literature on comparative tree biology/ architecture) is likely to not specifically understand what the authors are talking about.

> AR: We included the following examples the first time tree dimension was mentioned to avoid any confusion;
>
> CHANGE: [...] *scaling of tree dimensions –e.g. tree height or crown area– with trunk diameter.*

I found the paragraph (Lines 17-26) to be a little bit lacking in detail. First, I think the authors could include some examples of previous studies on the interspecific variation in tropical tree height and crown allometries. Secondly, the authors could include the information's on the relation between tree allometry and life-history traits. Lastly, the effects of species-specific or functional groups on the estimation of biomass should be developed in this section.

> AR:  We revised this paragraph to include references to previous work and summarized materials already included in the discussion to provide more background about the expected effect of functional traits on tree allometry;
>
> CHANGE: *Allometric studies of tropical trees have highlighted differences in growth and morphology that define distinct life history strategies (Clark and Clark, 1992; Poorter et al 1996). These differences contribute to species coexistence and play a key role in successional trajectories (Wright, 2002; Chazdon, 2014; Falster et al., 2017). Approaches pooling data across species inherently fail to recognize species heterogeneity in allometric scaling and limit the potential to identify and define plant functional groups. Pooling data across species also tends to over represent locally abundant species unless appropriate methods like hierarchical models are employed to account for unbalanced sampling.  Species differ systematically in allometric relationships, suggesting that these differences reflect underlying interspecific variation in life-history, physiology, morphology, and/or phylogeny (Westoby et al., 2002; Adler et al., 2014).  Hierarchical approaches based on functional traits can provide a useful approach for capturing this interspecific variation in tree allometry (Dietze et al., 2008; Iida et al., 2011). Several studies have found regeneration light requirements and/or wood density to be related to tree height and/or crown size across species, suggesting that these are good candidates for inclusion in a hierarchical model (Iida et al. 2012, 2014; Loubota Panzou et al. 2018; Poorter et al. 2006; Wright et al. 2010).*

Line 8: "the vegetation is moist tropical forest". Please specify the forest type: deciduousness forest or evergreen forest?

AR: *The forest is semideciduous.*

CHANGE: *tropical moist deciduous forest*

Line 10: ". . . with trunk diameter of 1 cm or larger . . ." What do you mean by "trunk larger"?

CHANGE: revised to *trunk diameter of at least 1 cm.*

Line 16: I propose the "tree measurements" rather than "allometric data".

CHANGE: Subsection title revised to: *2.2 Tree measurements*

Lines 16-25: this paragraph lacks of details on the data collection. I would like the authors' give more information on the compilation of these seven datasets.

AR: We edited the text to provide more details, and retain the reference to Table S1 in the Supplementary Material, which provides additional details on methods as well as sample sizes for each dataset. We also revised figure S1 in the supplement to include the average number of trees sampled per species and the extreme values as requested by R#2.

CHANGE: *We used a compilation of seven datasets collected in the BCI 50 ha plot and one dataset from the adjacent Gigante peninsula (see Table S1 for further details). The datasets cover different size classes and combine measurements made with different, albeit standard, methods. Depending on the dataset and tree size, tree heights were measured with a telescoping pole (smaller trees only), with a laser rangefinder using the sine or tangent method (Larjavaara and Muller-Landau 2013), or from the difference between phothotogrammetric estimation of canopy surface elevation and a digital elevation model obtained from airborne lidar (only fully sun-exposed trees). Crown areas were from ground-based measurements of crown radii, or from delimiting fully sun-exposed crowns in high-resolution aerial photos.*

Lines 5-12: the authors may add the names of species of low height or crown area and high height or crown area.

CHANGE: *The tallest species was* Dipteryx oleifera *(maximum height 57.4 m), and the largest crown areas were found in* Ceiba pentandra *(1404 m$^2$). Among big trees (DBH>80 cm),* Guazuma ulmifolia *presented the shortest tree (28.2 m), and* Poulsenia armata *the smallest crown area (179 m$^2$).*

Line 14: ". . . with dependence. . ." please reworded

CHANGE: Rephrased to: *The best tree height model combined a generalized Michaelis-Menten (gMM) function (Fig. 1a) with species specific parameters modeled as a linear function of sapling growth rates.*

Lines 16-17: "Past work suggests that mechanical resistance to self- or wind-loading cannot explain tree height allometries, as trees are generally much shorter for a given diameter than the limits based on static mechanical constraints (Niklas, 2007)". This is disconnected from your results.

AR: This sentence appears in a paragraph discussing mechanisms underlying tree height allometries, and it rules out one of the common hypothesis to explain tree height saturation with diameter.

Lines 12-13: Blanchard et al. (2016) study the variation inter-sites in tree allometry. Please see the reference Lines et al. (2012).

AR: Good point,

CHANGE: We have removed the reference to Blanchard et al. (2016) from this sentence on interspecific variation in tree allometry, and added the reference to Lines et al (2012).

Lines 30-31: This last sentence isn't necessary.

CHANGE: Removed.

This Fig.4 isn't necessary.

AR: We see Figure 4 as a useful illustration of the mechanisms underlying the differences in plot-level biomass estimates, and choose to retain it.

*References*

Prentice, I. C., Cramer,W., Harrison, S. P., Leemans, R., Monserud, R. A., and Solomon, A. M.: A Global Biome Model Based on Plant Physiology and Dominance, Soil Properties and Climate, J. Biogeogr., 19, 117–134, 1992.